EMBO
Molecular Medicine

# Glia inflammation and cell death pathways drive disease progression in preclinical and early AD

Marcel S Woo [1,2,3✉], Joseph Therriault [1,4,5], Seyyed Ali Hosseini[1,4,5], Yi-Ting Wang[1,4,5], Arthur C Macedo[1,4,5], Nesrine Rahmouni[1,4,5], Étienne Aumont[1,4,5], Stijn Servaes [1,4,5], Cécile Tissot [1,6], Jaime Fernandez-Arias[1,4,5], Lydia Trudel[1,4,5], Brandon Hall[1,4,5], Gleb Bezgin[1,4,5], Kely Quispialaya-Socualaya[1,4,5], Marina Goncalves[1,4,5], Tevy Chan [1,4,5], Jenna Stevenson[1,4,5], Yansheng Zheng[1,4,5], Stuart Mitchell [1,4,5], Robert Hopewell [1,4,5], Ilaria Pola[7], Kubra Tan[7], Guglielmo Di Molfetta[7], Firoza Z Lussier [8,9], Gassan Massarweh[5], Paolo Vitali[1,4,5], Jean-Paul Soucy[5], Serge Gauthier[1,4,5], Nicholas J Ashton [7,10,11,12], Kaj Blennow [7,13], Tharick A Pascoal[8,9], Henrik Zetterberg [7,13,14,15,16,17], Andréa L Benedet[7,19] & Pedro Rosa-Neto [1,4,5,18,19✉]

## Abstract

Accumulation of amyloid-β (Aβ) and neurofibrillary tangles (NFTs) are followed by the activation of glia cells and infiltration of peripheral immune cells that collectively accelerate neurodegeneration in preclinical AD models. Yet, the role of neuroinflammation for neuronal injury and disease progression in preclinical and early symptomatic AD remains elusive. Here, we combined multiplexed immunoassays and SomaScan proteomics of the cerebrospinal fluid (CSF) with MRI and PET brain imaging of people across the AD continuum to identify pathways that are associated with AD progression. Unbiased clustering revealed that glia-mediated inflammation, activation of cell death pathways (CDPs) and synaptic pathologies were among the earliest Aβ-induced changes, and were associated with disease progression in preclinical AD. Mediation analysis revealed that activation of CDPs were decisive drivers of inflammation in early symptomatic AD. The cycle of glia-mediated neuroinflammation and neuronal injury characterizes preclinical AD and has implications for novel treatment approaches.

**Keywords** Alzheimer's Disease; Glia Inflammation; Cell Death; Proteomics; Biomarker
**Subject Category** Neuroscience

## Introduction

The core pathologies of Alzheimer's disease (AD), amyloid-β (Aβ) and neurofibrillary tangles (NFTs) (Therriault et al, 2022; Pascoal et al, 2020) aggregations, are accompanied by chronic inflammation in the central nervous system (CNS) (Heneka et al, 2025a). This concept has been supported by increased concentrations of inflammatory cytokines in the brain and CSF of people along the AD continuum, as well as identification of several activated glia states in postmortem histopathological and single-cell-sequencing studies (Gate et al, 2020; Mathys et al, 2024; Sung et al, 2023; Hong et al, 2016; Chen et al, 2023; Green et al, 2024). Furthermore, imaging studies in people across the AD continuum showed that tau propagation follows the spread of activated microglia (Pascoal et al, 2021) and immunophenotyping of the CSF from people with AD revealed an increased number of activated peripheral immune cells (Berriat et al, 2023; Gate et al, 2020) providing further evidence for the interplay between the adaptive and innate immune system in AD (Heneka et al, 2025a). Mouse studies have provided additional evidence that Aβ and NFTs lead to the activation of glia

[1]Translational Neuroimaging Laboratory, The McGill University Research Centre for Studies in Aging, McConnell Brain Imaging Centre (BIC), Montreal Neurological Institute, Montréal, QC, Canada. [2]Translational Neurodegeneration Laboratory, Department of Neurology, University Medical Centre Hamburg Eppendorf, Hamburg, Germany. [3]Institute of Neuroimmunology and Multiple Sclerosis, University Medical Centre Hamburg Eppendorf, Hamburg, Germany. [4]Department of Neurology and Neurosurgery, McGill University, Montréal, QC, Canada. [5]Montreal Neurological Institute, Montréal, QC, Canada. [6]Lawrence Berkeley National Laboratory, 1 Cyclotron Rd, Berkeley, CA, USA. [7]Department of Psychiatry and Neurochemistry, Institute of Neuroscience and Physiology, The Sahlgrenska Academy, University of Gothenburg, Mölndal, Sweden. [8]Department Psychiatry, School of Medicine, University of Pittsburgh, Pittsburgh, PA, USA. [9]Department of Neurology, School of Medicine, University of Pittsburgh, Pittsburgh, PA, USA. [10]Institute of Psychiatry, Psychology and Neuroscience Maurice Wohl Institute Clinical, King's College London, Neuroscience Institute London, London, UK. [11]Banner Alzheimer's Institute and University of Arizona, Phoenix, AZ, USA. [12]Banner Sun Health Research Institute, Sun City, AZ 85351, USA. [13]Clinical Neurochemistry Laboratory, Sahlgrenska University Hospital, Mölndal, Sweden. [14]Department of Neurodegenerative Disease, UCL Institute of Neurology, Queen Square, London, United Kingdom. [15]UK Dementia Research Institute at UCL, London, UK. [16]Hong Kong Center for Neurodegenerative Diseases, Clear Water Bay, Hong Kong, China. [17]Wisconsin Alzheimer's Disease Research Center, University of Wisconsin School of Medicine and Public Health, University of Wisconsin-Madison, Madison, WI, USA. [18]Peter O'Donnell Jr. Brain Institute (OBI), University of Texas Southwestern Medical Centre (UTSW), Dallas, TX, USA. [19]These authors jointly supervised this work: Andréa L Benedet, Pedro Rosa-Neto. ✉E-mail: m.woo@uke.de; pedro.rosa@mcgill.ca

cells, which induce the infiltration of peripheral immune cells. This leads to secretion of pro-inflammatory cytokines or other toxic substrates that maintain a chronic inflammatory environment, accelerating neuronal injury in mouse models and in cell culture (Jorfi et al, 2023; Chen et al, 2023; Mancuso et al, 2024). Furthermore, inflammatory cell death of microglia, pyroptosis, strongly promotes Aβ-accumulation, providing mechanistic evidence for an involvement of inflammation in the earliest AD stages (Venegas et al, 2017; Heneka, 2017). At the same time, several studies have shown a protective role for microglia by reducing Aβ accumulation and NFT spreading. Genome-wide association studies identified genetic risk factors that reduce the phagocytic capacity of microglia, and increase the risk of developing AD. Furthermore, boosting phagocytosis is protective in AD mouse models by limiting Aβ plaques and NFT aggregation (Bellenguez et al, 2022; van Lengerich et al, 2023; Schlepckow et al, 2023; Pereira et al, 2022), underscoring the multifaceted role of inflammation for AD progression. However, it is unclear whether neuroinflammation appears secondary to Aβ and NFT accumulation, during later disease stages, or whether it might be an earlier process already present in preclinical stages of AD.

To address this knowledge gap, we deployed a NUcleic acid-Linked Immuno-Sandwich Assay (NULISA) (Feng et al, 2023) for multiplexed immunoassay profiling, and SomaScan proteomics of a longitudinal cohort to identify biological pathways associated with disease progression in people with cognitive impairment (CI) or without (CU). Our hypothesis is that glia-mediated inflammation is associated with disease progression and neuronal injury in preclinical AD.

# Results

## Patient characteristics

We aimed to identify biological signatures that define different stages of cognitive decline in AD progression. To separate AD-

We included 382 individuals from the Translational Biomarkers in Aging and Dementia (TRIAD) cohort, where the NULISA CNS panel was measured. This encompassed 32 cognitively unimpaired individuals younger than 30 years of age (CUY, mean age 23 years), 154 CU older than 30 years (mean age 68 years), 39 people with mild cognitive impairment due to AD (MCI, mean 71 years), 50 people with dementia due to AD (ADD, mean age 66 years) and 107 people with other neurological diseases (OND, mean age 68 years). Overall, 221 individuals were female (57%; 20 CUY, 99 CU, 20 MCI, 19 ADD, 63 OND). Longitudinal data were available for 146 individuals (81 females, mean age 67 years) with a mean follow-up of 26 months. The demographics are summarized in Appendix Table S1.

Additionally, we included 430 individuals from the Alzheimer's Disease Neuroimaging Initiative (ADNI), where the 7K SomaScan assay was performed with the CSF. This encompassed 114 CU older than 30 years (mean age 76 years), 213 people with MCI (mean age 73 years), and 103 people with ADD (mean age 75 years). SomaScan data were only available for one time point. The demographics for ADNI are summarized in Appendix Table S2.

## Distinct protein signatures define clinical progression in AD

We aimed to identify biological signatures that define different stages of cognitive decline in AD progression. To separate AD-

specific signatures from aging, we first investigated age-dependent changes in the CSF proteome. This was only possible in TRIAD because no CUY individuals were available in ADNI. We analyzed the CSF of all participants in TRIAD using the NULISA (Feng et al, 2023) CNS panel, which has attomolar sensitivity to detect low-abundance proteins. We contrasted the CSF proteome of CUY against CU, MCI, and ADD groups separately (Appendix Fig. S1A–C) and applied gene ontology analysis of biological pathways (GO-BP) to define protein signatures of the differentially abundant proteins that change across AD stages (Appendix Fig. S1A–C). In line with findings from other studies (Oh et al, 2023; López-Otín et al, 2023), these comparisons revealed a significant enrichment of immune signaling, glia activation, calcium signaling pathways as well as specific cytokine signaling pathways like those of interleukin (IL)-1β, tumor necrosis factor α (TNFα) and interferon γ (IFNγ) (Appendix Fig. S1D,E). To separate age-driven and Aβ-specific effects, we performed differential abundance analyses between CU with negative Aβ status measured by PET (A−) and CUY A− to determine protein changes linked to aging independent of Aβ (Appendix Fig. S1F), as well as between CI with positive Aβ status measured by PET (A+) and CU A− (Fig. 1A) to isolate the Aβ-specific changes. GO-BP term enrichment analysis revealed that most inflammatory pathways were driven by ageing (Appendix Fig. S1G) while CI A+ individuals were characterized by disturbances in metabolism, synaptic signaling, and glia activation (Fig. 1B–D) as determined by GO-BP analysis (the top 20 GO terms are shown in the Appendix Table S3).

To validate these findings, we performed a differential abundance analysis between the CI A+ and CU A− ADNI participants using the 7K SomaScan proteomics data (Fig. 1E). Again, we performed GO-BP analysis (the top 20 GO terms are shown in the Appendix Table S3) and identified an increase of proteins associated with metabolism, synaptic signaling, and glia activation. Notably, we additionally detected an increased protein folding signature, another important pathway in AD pathophysiology (Fig. 1F–H). Thus, we identified common disturbed pathways in people with clinical AD in two independent AD continuum cohorts.

In the next step, we investigated the preclinical AD stage by comparing the CSF proteome of CU A− against CU A+, excluding the CUY. This analysis in TRIAD revealed an alteration of eight proteins in CU A+ that were associated with synaptic organization and metabolic processes (Fig. 1I,J). Since we observed that disturbed synaptic signaling and glia cell activation were significantly enriched in the CSF proteomes of CI A+ (Fig. 1A–H), we inquired whether Aβ already induced these pathways in CU cases. To test this hypothesis, we generated scores for each participant considering all proteins defining the different biological processes. We detected larger anomalies of protein concentration for proteins linked to synaptic signaling and glia cell activation in CU A+ in comparison to CU A− (Fig. 1K,L). In contrast, GFAP was not significantly upregulated in the CSF of CI A+ and CU A+ in comparison to CU A− in accordance with previous reports (Pereira et al, 2021; Benedet et al, 2021) (Appendix Fig. S1H,I), underlining the advantage of using multi-protein signatures in comparison to individual protein anomalies. Of note, we detected significantly elevated GFAP levels in the plasma of CI A+ and CU A+ in comparison to CU A− (Appendix Fig. S1J,K) further supporting

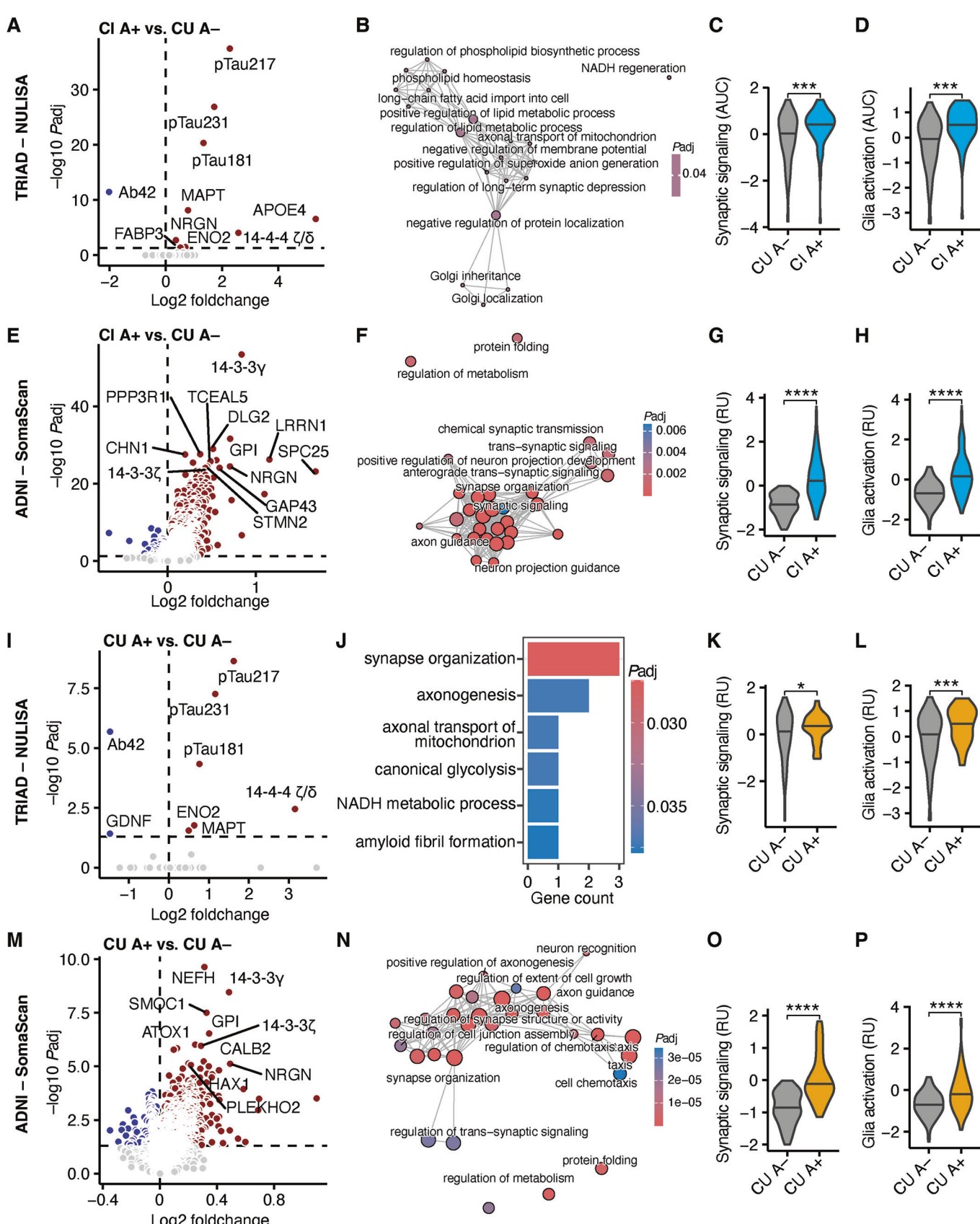

**Figure 1.    AD stages have distinct protein signatures.**

(A) Volcano plot of CSF proteins in cognitively impaired with positive Aβ status (CI A+, $n = 73$) and cognitively unimpaired with negative Aβ status (CU A−, $n = 160$; B) in TRIAD using NULISA. Differentially abundant proteins are labeled with colors. (B) Emapplot of biological process gene ontology analysis of differentially abundant CSF proteins in CI A+ vs. CU A− in TRIAD. (C, D) Area under the curve analysis of the GO terms "synaptic signaling" (C; $P = 0.00073$) and "glia activation" (D; $P = 3.3 \times 10^{-8}$) in CI A+ vs. CU A− in TRIAD. Z-scores are shown in relative units (RU). (E) Volcano plot of CSF proteins in cognitively impaired with positive Aβ status (CI A+, $n = 316$) and cognitively unimpaired with negative Aβ status (CU A−, $n = 68$; B) in ADNI using 7K SomaScan. Differentially abundant proteins are labeled with colors. (F) Emapplot of biological process gene ontology analysis of differentially abundant CSF proteins in CI A+ vs. CU A− in ADNI. (G, H) Area under the curve analysis of the GO terms "synaptic signaling" (G; $P = 4.5 \times 10^{-32}$) and "glia activation" (H; $P = 8.2 \times 10^{-20}$) in CI A+ vs. CU A− in ADNI. Z-scores are shown in relative units (RU). (I) Volcano plot of CSF proteins in CU with positive Aβ status (CU A+, $n = 29$) in comparison to CU with negative Aβ status (CU A−, $n = 160$) in TRIAD using NULISA. Differentially abundant proteins are labeled with colors. (J) GO term enrichment analysis of differentially abundant CSF proteins in CU A+ vs. CU A− in TRIAD. (K, L) Area under the curve analysis of synaptic signaling (K; $P = 0.029$) and glia activation (L; $P = 0.0007$) pathways in CU A+ vs. CU A− in TRIAD. Z-scores are shown in relative units (RU). (M) Volcano plot of CSF proteins in CU with positive Aβ status (CU A+, $n = 46$) in comparison to CU with negative Aβ status (CU A−, $n = 68$) in ADNI using 7K SomaScan. Differentially abundant proteins are labeled with colors. (N) GO term enrichment analysis of differentially abundant CSF proteins in CU A+ vs. CU A− in ADNI. (O, P) Area under the curve analysis of synaptic signaling (O; $P = 3.7 \times 10^{-10}$) and glia activation (P; $P = 0.00025$) pathways in CU A+ vs. CU A− in ADNI. Z-scores are shown in relative units (RU). If not stated otherwise, a two-tailed $t$-test with FDR-correction were used for group comparisons. $*P < 0.05$ and $***P < 0.001$. Source data are available online for this figure.

that NULISA recapitulates previous findings in other cohorts and different methods.

We repeated the differential abundance analysis between CU A+ and CU A− in ADNI (Fig. 1M) and performed GO-BP with the differentially more abundant proteins. Similar to our analysis in TRIAD, we found an enrichment of proteins that are associated with synaptic signaling, metabolism, and glia activation. In addition, we identified increased protein folding and cell chemotaxis signatures (Fig. 1N–P; the top 20 GO terms are shown in the Appendix Table S3). The striking similarities in the CSF proteomics changes in preclinical and clinical AD in comparison to CU A− in two independent cohorts let us conclude that Aβ accumulation leads to changes in glia activation and synaptic signaling already in preclinical AD stages.

## Clinical biomarkers are associated with different biological processes

In the next step, we aimed to identify biological processes that were gradually associated with AD hallmarks and might not be captured by differential abundance analyses. Therefore, we tested the correlation between each protein and neocortical Aβ load, tau load in different Braak stages, hippocampal volume, and cognitive performance (Fig. 2A,B) and used unbiased clustering to identify six protein clusters (Appendix Fig. S2A–C). This analysis was only performed in TRIAD because no tau-PET data were available for the included ADNI participants. To assess the overall association of the clusters with different AD hallmarks, we averaged the correlation coefficients of each of the proteins of each cluster. Cluster 6 included phosphorylated tau (p-tau) isoforms p-tau181, p-tau217, p-tau231, and total tau, and cluster 3 included only Aβ42 due to its strong negative association with the AD hallmarks (Fig. 2C). Cluster 1 showed the strongest association with Aβ accumulation but also significantly correlated with tau load and cognitive performance. To characterize the proteins in each cluster, we performed GO-BP analysis. Cluster 1 included proteins that are associated with the cellular response to Aβ as well as glia activation, synaptic and calcium signaling (Fig. 2D). Additionally, cluster 5 was strongly correlated with tau load in Braak I and II areas as well as neocortical Aβ load. Of note, this cluster included proteins involved with neuronal pathologies, like activation of cell death pathways (CDPs), and the regulation of the neuronal cytoskeleton and mitochondrial transport (Appendix Fig. S2D). Clusters 2 and 4

that consisted of developmental processes and IL1β signaling pathways, did not show significant correlations with the AD hallmarks (Appendix Fig. S2E,F). Thus, we identified two clusters that were highly correlated with AD hallmarks and consisted of proteins that were associated with neuroinflammation, glia activation, and neuronal pathologies.

In the next step, we investigated intraindividual differences in anomalies of the different pathways and their association with disease progression. Therefore, we compiled scores for the following GO terms that were included in clusters 1 and 5 of our previous analysis and were significantly correlated with AD hallmarks (Fig. 2): neuroinflammation, glia activation, cell death pathways (CDPs), mitochondrial axonal transport, and synaptic signaling (the defining proteins are shown in the Appendix Table S4). Notably, the neuroinflammation score included all proteins of the glia activation score and therefore, represented a mother term for glia activation and other inflammatory processes. The protein resolution did not allow further differentiation of glia activation states or subtypes. First, we compared the scores across the different diagnostic groups. In ADNI, we detected a significant increase in all scores in MCI and ADD in comparison to CU (Appendix Fig. S3A–E). In TRIAD, we detected a significant increase in CU, MCI, and ADD in comparison to CUY, as well as an additional increase in MCI and ADD in comparison to CU. Notably, only mitochondrial axonal transport was additionally increased in ADD in comparison to MCI (Appendix Fig. S3F–J). Next, we performed correlation analysis of these scores with neocortical Aβ load, tau load, hippocampal volume, and cognitive performance. We identified similar patterns for neuroinflammation, glia activation, CDPs, and synaptic signaling, which significantly correlated with neocortical Aβ load, and tau load in Braak I-III but not with cognitive performance (Appendix Fig. S3K–O). In contrast, axonal mitochondrial transport showed only a weakly significant correlation with Aβ and tau load but was significantly associated with cognitive performance independent of hippocampal volume. Thus, our data show that specific pathological processes are associated with the biological and clinical phenotypes in AD.

## Inflammation, glia activation, and cell death pathways predict disease progression

Next, our goal was to assess whether these processes were associated with disease progression in early AD stages. Therefore,

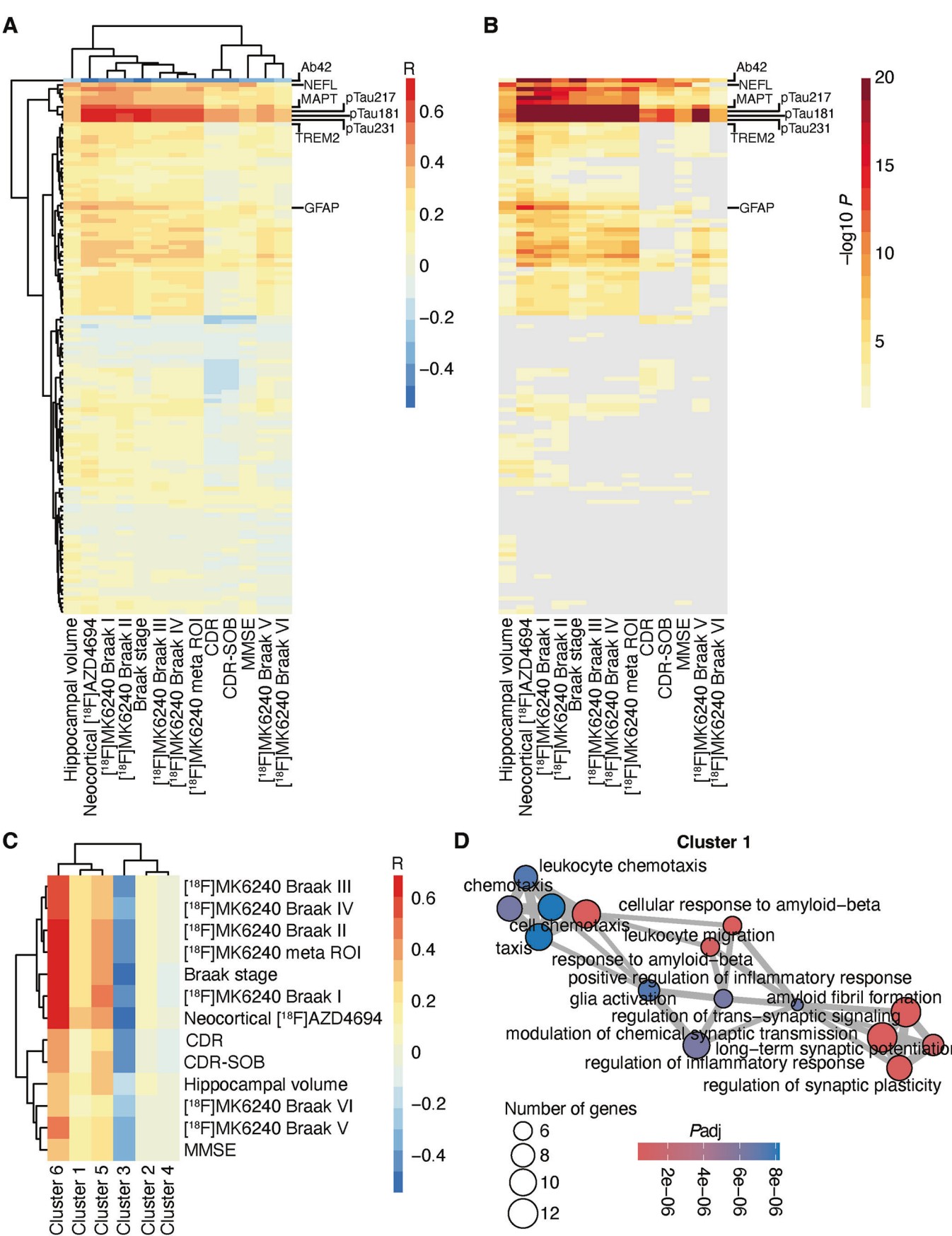

**Figure 2.   Distinct protein clusters are associated with AD hallmarks.**

(A) Heatmap of Spearman correlation coefficients of all measured proteins (y-axis) and hippocampal volume, neocortical [¹⁸F]AZD4694 SUVR, [¹⁸F]MK6240 SUVR in Braak I–VI and meta-ROI, Braak stages, mini-mental state examination (MMSE), clinical dementia rating (CDR), and CDR-SOB (x-axis). The correlation coefficients for hippocampal volume and cognitive scores were inverted. Hierarchical clustering of rows and columns was used. (B) Significance of correlation analysis. The heatmap has the same order as shown in A. Spearman correlation P value >0.05 is shown as gray. (C) Heatmap of individual clusters with AD hallmarks (same as in A) using the averaged R for each cluster. Hierarchical clustering for rows and columns was used. (D) Gene ontology analysis of all proteins that were included in cluster 1 was calculated by FDR-corrected overrepresentation analysis. Size shows number of genes, color shows adjusted P value of enrichment. Source data are available online for this figure.

we separated the individuals according to the A/T/N allowing us to analyze 68 A−T−N−, 77 A+T−N−, 145 A+T+N−, and 140 A+T+N+ individuals from ADNI, and 171 A−T−N−, 33 A +T−N−, 35 A+T+N−, and 25 A+T+N+ TRIAD participants with PET and MRI images available (demographics separated by A/T/N are provided in Appendix Table S5 for ADNI, and Appendix Table S6 for TRIAD). We detected a gradual increase of neuroinflammation, glia activation, CDPs, mitochondrial axonal transport, and synaptic signaling across the A/T stages in both cohorts. Markers of neurodegeneration did not correlate with any of the pathways (Fig. 3A–H and Appendix Fig. S4A,B). Subsequently, we tested the correlation of these pathways in TRIAD with the AD hallmarks within the individual A/T/N groups in all individuals (Fig. 3I) or individuals older than 50 years to exclude aging effects (Appendix Fig. S4C). Intriguingly, all pathways were significantly associated with cortical Aβ load in A−T−N− but not in the other A/T/N combinations (Fig. 3J–M; Appendix Fig. S4D). Additionally, synaptic signaling, neuroinflammation, CDPs, and glia activation pathways were significantly associated with tau load in Braak IV and V in A+T−N− cases but not in the other A/T/N groups (Fig. 3N–Q; Appendix Fig. S4E–G). When including all individuals, the pathways were significantly associated with hippocampal volume in A−T−N−. However, only mitochondrial axonal transport was still significantly associated with hippocampal volume when including individuals older than 50 years (Appendix Fig. S4H), underlining that the association with hippocampal volume was likely age-related. We concluded that these proteomic CSF signatures were associated with AD hallmarks in the early disease stages.

To test whether activation of these pathways precedes Aβ and tau accumulation and might be associated with early AD progression, we looked at our longitudinal data in TRIAD and used p-tau217 as a biomarker for AD progression. We compared 84 CU A- (47 female, mean age 64 years, mean follow-up 25 months) with 22 CU A+ (11 female, mean age 70 years, mean follow-up 25 months), and four CI A− (four female, mean age 66 years, mean follow-up 15 months) with 36 CI A+ (five female, mean age 72 years, mean follow-up 27 months). As expected, we detected an increase of p-tau217 in CU A+ in comparison to CU A− (Appendix Fig. S5A). Similarly, the CSF signatures for glia activation, neuroinflammation, CDP activation, mitochondrial axonal transport deficits, and synaptic signaling significantly increased over time in CU A+ but not in CU A− (Fig. 4A–E). The interaction between the longitudinal increase and Aβ-status was significant for glia activation, neuroinflammation, CDP activation, and mitochondrial axonal transport deficits, underlining that these pathways were stronger associated with progression in preclinical AD than physiological aging in the observed time

period. We further found a strong correlation between the yearly change of p-tau217 and these biological processes in CU (Fig. 4F±J), but we did not find a different progression of anomalies of these pathways between CI A+ and CI A− (Appendix Fig. S5B–G). Thus, longitudinal disease progression in preclinical AD is associated with CSF signatures of neuroinflammation, glia inflammation, activation of cell death pathways, and neuronal pathologies like disturbances in mitochondrial axonal transport and synaptic signaling.

## Neuroinflammation feeds a vicious cycle that drives early AD

After discovering pathways that were associated with early AD progression, we aimed to uncover possible promoters of neuroinflammation. We focused on early AD and analyzed A−T− and A+T− individuals and tested whether activation of glia and CDPs mediate the association between tau load and neuroinflammation. This analysis was only done in TRIAD because no tau-PET data were available for the included ADNI participants. Although there was a significant total effect in these models, we found no significant direct effect of tau load on neuroinflammation. Instead, there was a significant association between the activation of CDPs (Fig. 5A) and glia (Fig. 5B) with tau load and neuroinflammation. Activation of CDPs and glia were significant mediators of NFT-dependent neuroinflammation. This was also the case for tau load in Braak stages IV and V in A+T+ individuals (Appendix Fig. S6A,B), indicating that activation of CDPs and glia activation are important contributors to tau-associated inflammation in early and later disease stages.

Next, we assessed how AD pathology measured by CSF p-tau217 in TRIAD (Appendix Fig. S6C), and CSF p-tau181 in ADNI, might lead to neuroinflammation in A−T− and A+T−. Similar to tau load, we did not detect a significant direct effect of p-tau217 and p-tau181 on neuroinflammation. However, the total effect was significant due to strong mediation by CDPs in TRIAD (Fig. 5C) and ADNI (Fig. 5D), further supporting the importance of PCD activation for the initial phase of AD (Heneka, 2017; Venegas et al, 2017). At the same time, neuroinflammation was significantly associated with CDP activation, which was additionally mediated by synaptic signaling in TRIAD (Fig. 5E) and ADNI (Fig. 5F). The CSF proteomic signature of mitochondrial axonal transport was a significant mediator in TRIAD (Appendix Fig. S6D) but not in ADNI (Appendix Fig. S6E), suggesting that neuroinflammation promotes CDP activation by perturbing the synaptic homeostasis. Thus, our data provide evidence that disease progression in early AD is driven by activation of CDPs that results in glia activation and neuroinflammation, which subsequently lead to further

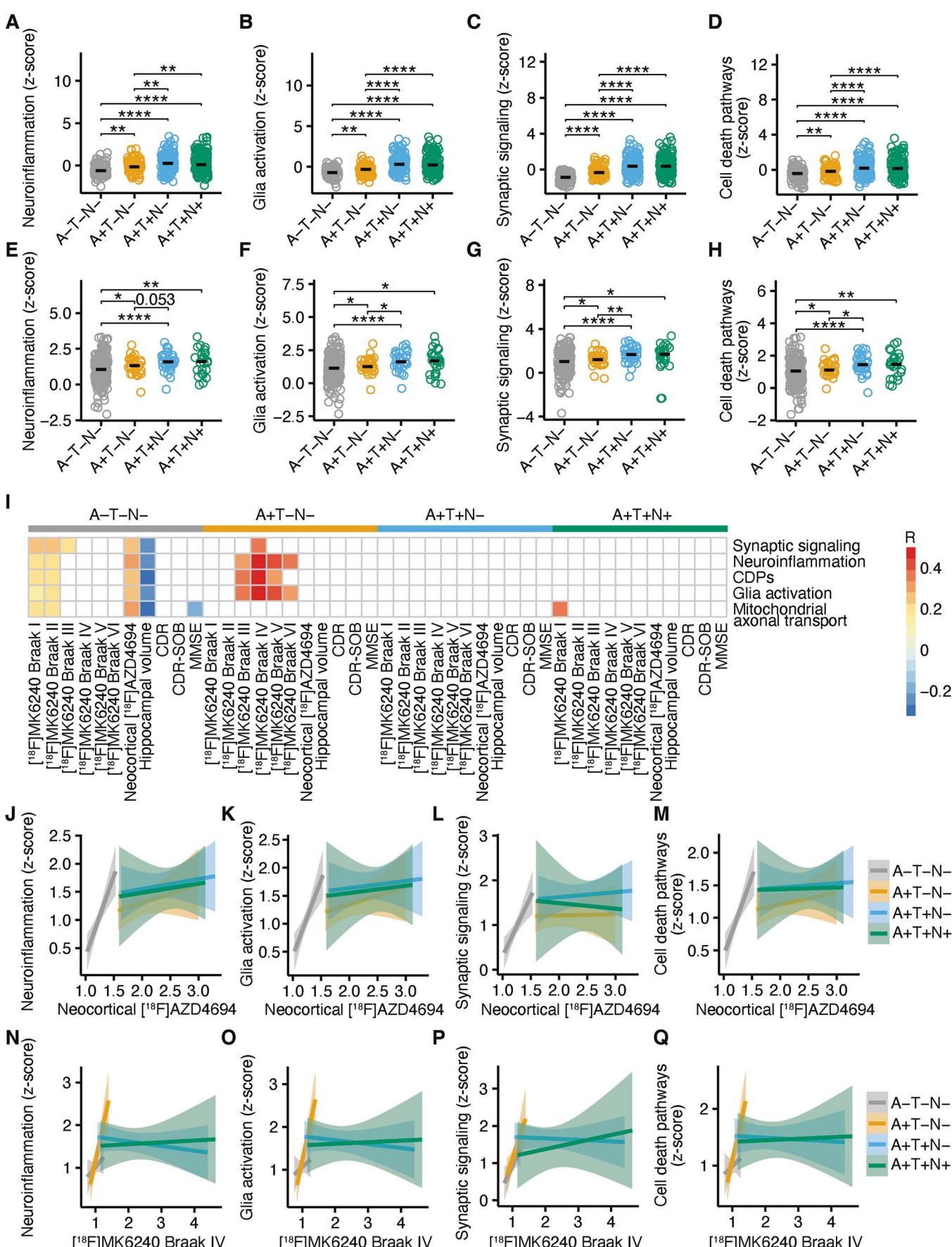

**Figure 3.  Glia activation, neuroinflammation, and cell death pathways precede Aβ and tau aggregation.**

(A–D) Z-score of biological pathways "Neuroinflammation" (A; A+T+N+ vs. A−T−N−, $P=1.6 \times 10^{-7}$; A+T+N+ vs. A+T−N−, $P=0.0088$; A+T+N+ vs. A+T+N−, $P=0.73$ (n.s.); A−T−N− vs. A+T−N−, $P=0.0053$; A−T−N− vs. A+T+N−, $P=6.2 \times 10^{-9}$; A+T−N− vs. A+T+N−, $P=0.0019$), "Glia activation" (B; A+T+N+ vs. A−T−N−, $P=6.4 \times 10^{-17}$; A+T+N+ vs. A+T−N−, $P=2.2 \times 10^{-10}$; A+T+N+ vs. A+T+N−, $P=0.62$; A−T−N− vs. A+T−N−, $P=0.0035$; A−T−N− vs. A+T+N−, $P=7.1 \times 10^{-20}$; A+T−N− vs. A+T+N−, $P=9.0 \times 10^{-13}$), "Synaptic signaling" (C; A+T+N+ vs. A−T−N−, $P=2.4 \times 10^{-27}$; A+T+N+ vs. A+T−N−, $P=3.7 \times 10^{-11}$; A+T+N+ vs. A+T+N−, $P=0.77$; A−T−N− vs. A+T−N−, $P=6.1 \times 10^{-9}$; A−T−N− vs. A+T+N−, $P=5.8 \times 10^{-28}$; A+T−N− vs. A+T+N−, $P=5.3 \times 10^{-11}$), and "Cell death pathways" (D; A+T+N+ vs. A−T−N−, $P=2.2 \times 10^{-9}$; A+T+N+ vs. A+T−N−, $P=5.2 \times 10^{-5}$; A+T+N+ vs. A+T+N−, $P=0.59$; A−T−N− vs. A+T−N−, $P=0.0087$; A−T−N− vs. A+T+N−, $P=3.0 \times 10^{-11}$; A+T−N− vs. A+T+N−, $P=1.6 \times 10^{-6}$) in A−T−N− (n=68), A+T−N− (n=77), A+T+N− (n=145), and A+T+N+ (n=140) in ADNI. An unpaired t-test with FDR-correction was used for statistical comparisons. Individuals with a z-score >5 were excluded as outliers. (E–H) Z-score of biological pathways "Neuroinflammation" (E; A−T−N− vs. A+T−N−, $P=0.031$; A−T−N− vs. A+T+N−, $P=3.4 \times 10^{-5}$; A−T−N− vs. A+T+N+, $P=0.0069$; A+T−N− vs. A+T+N−, $P=0.053$; A+T−N− vs. A+T+N+, $P=0.21$; A+T+N− vs. A+T+N+, $P=0.83$), "Glia activation" (F); A−T−N− vs. A+T−N−, $P=0.030$; A−T−N− vs. A+T+N−, $P=3.9 \times 10^{-5}$; A−T−N− vs. A+T+N+, $P=0.011$; A+T−N− vs. A+T+N−, $P=0.040$; A+T−N− vs. A+T+N+, $P=0.23$; A+T+N− vs. A+T+N+, $P=0.78$) "Synaptic signaling" (G); A−T−N− vs. A+T−N−, $P=0.043$; A−T−N− vs. A+T+N−, $P=3.0 \times 10^{-6}$; A−T−N− vs. A+T+N+, $P=0.083$; A+T−N− vs. A+T+N−, $P=0.0084$; A+T−N− vs. A+T+N+, $P=0.47$; A+T+N− vs. A+T+N+, $P=0.43$), and "Cell death pathways" (H; A−T−N− vs. A+T−N−, $P=0.043$; A−T−N− vs. A+T+N−, $P=7.2 \times 10^{-5}$; A−T−N− vs. A+T+N+, $P=0.0079$; A+T−N− vs. A+T+N−, $P=0.032$; A+T−N− vs. A+T+N+, $P=0.16$; A+T+N− vs. A+T+N+, $P=0.84$) in A−T−N− (n=171), A+T−N− (n=33), A+T+N− (n=35), and A+T+N+ (n=25) in TRIAD. An unpaired t-test with FDR-correction was used for statistical comparisons. Individuals with a z-score >5 were excluded as outliers. (I) Spearman correlation coefficients separated by A/T/N of the biological pathways "Neuroinflammation", "Glia activation", "Mitochondrial axonal transport", "Cell death pathways" (CDPs), and synaptic signaling with hippocampal volume, neocortical [18F]AZD4694 SUVR, [18F]MK6240 SUVR in Braak I-VI and meta-ROI, Braak stages, mini-mental state examination (MMSE), clinical dementia rating (CDR) and CDR-SOB in TRIAD. White tiles represent P value >0.05. (J–M) Regression lines and 95% confidence intervals of the association between "Neuroinflammation" (J), "Glia activation" (K) "Synaptic signaling" (L), and "Cell death pathways" (M) and neocortical [18F]AZD4694 SUVR separated by A/T/N in TRIAD. Significance and correlation coefficients are shown in 3I. (N–Q) Regression lines and 95% confidence intervals of the association between "Neuroinflammation" (N), "Glia activation" (O) "Synaptic signaling" (P), and "Cell death pathways" (Q) and Braak IV [18F]MK6240 SUVR separated by A/T/N in TRIAD. Significance and correlation coefficients are shown in 3I. *$P < 0.05$, **$P < 0.01$, ***$P < 0.001$, ****$P < 0.0001$. Source data are available online for this figure.

neuronal disturbances, maintaining a vicious cycle of continuous CNS inflammation (Fig. 5G).

## Discussion

In this paper, we characterized vulnerable pathways associated with disease progression in preclinical and early symptomatic AD by using NULISA (Feng et al, 2023) multiplex immunoassay and SomaScan proteomics technologies in CSF samples. NULISA has a detection range to the attomole levels, which allows the quantification of low-abundance proteins like cytokines or chemokines to examine AD-related differences in biomarker levels. By combining the protein quantification data with several imaging modalities and clinical assessments, we identified pathways that were associated, or not, with disease progression. Notably, we detected that immune pathways like chemotaxis and immune cell migration were associated with aging. This is in line with several omics studies that identified immune signatures and infiltrates in brains from aged subjects (Dulken et al, 2019; Oh et al, 2023; Almanzar et al, 2020; Lehallier et al, 2019). Differential abundance analysis revealed in two independent cohorts an additional Aβ-specific disturbance of metabolic pathways, synaptic signaling, and glia activation in CI and CU subjects. This further underlines the importance of metabolic dysregulation for AD pathogenesis, which is in line with mechanistic studies in preclinical models that highlight neuronal and glia metabolism as attractive treatment targets in AD (Minhas et al, 2024; Byrns et al, 2024).

Intriguingly, synaptic signaling pathways were altered in CU A+ and CI A+ in comparison to A− individuals, adding further evidence to their involvement in early symptomatic AD (Oh et al, 2025). This notion is further supported by postmortem studies that identified a redistribution of highly calcium-permeable ionotropic glutamate receptors in AD to extrasynaptic sites, which favors glutamate excitotoxicity (Escamilla et al, 2024), a calcium-

dependent form of cell death that is elicited by neuronal hyperexcitation. Furthermore, a large CSF proteomics dataset of risk carriers for autosomal dominant AD (ADAD) identified excitotoxicity as an important contributor to disease progression (Shen et al, 2024). Notably, in the ADAD cohort as well as in our study, disturbances in synaptic signaling were already detectable and associated with disease progression in CU A+ (Shen et al, 2024). Therefore, this study advocates the need for early identification of Aβ+ individuals who are at risk of developing AD due to the early emergence of neuronal injury.

Also, a CSF signature of glia activation was already present in CU A+ in comparison to CU A−, and glia activation and neuroinflammation were associated with disease progression only in the presence of Aβ. However, our and other cross-sectional studies did not find differences in CSF GFAP in CU A+ compared to CU A− (Benedet et al, 2021; Pereira et al, 2021), underlining the strength of our approach to detect disease-driving pathways not detected by single biomarker analysis in early AD. Mechanistic studies in mice and biomarker studies in humans have clearly demonstrated the importance of glia activation and neuroinflammation for disease progression in different AD stages for tau spreading, Aβ accumulation and neurodegeneration (Serrano-Pozo et al, 2024; Habib et al, 2020; Shen et al, 2019; Heneka et al, 2025b; Heneka, 2017; Venegas et al, 2017; Foley et al, 2024). In line with a study that demonstrated that Aβ-induced tau accumulation requires reactive astrogliosis (Bellaver et al, 2023), we show that glia-mediated neuroinflammation is longitudinally associated with disease progression in the CU A+ preclinical stage. Similarly, CDP activation increased with longitudinal AD progression in CU A+, which is in line with cross-sectional studies that identified a linear increase of CDP activation along the AD continuum (Ali et al, 2025).

Our mediation analysis revealed that in early AD, activation of CDPs and glia were important mediators of Aβ-induced neuroinflammation. However, it is still unclear in which cell types CDPs are

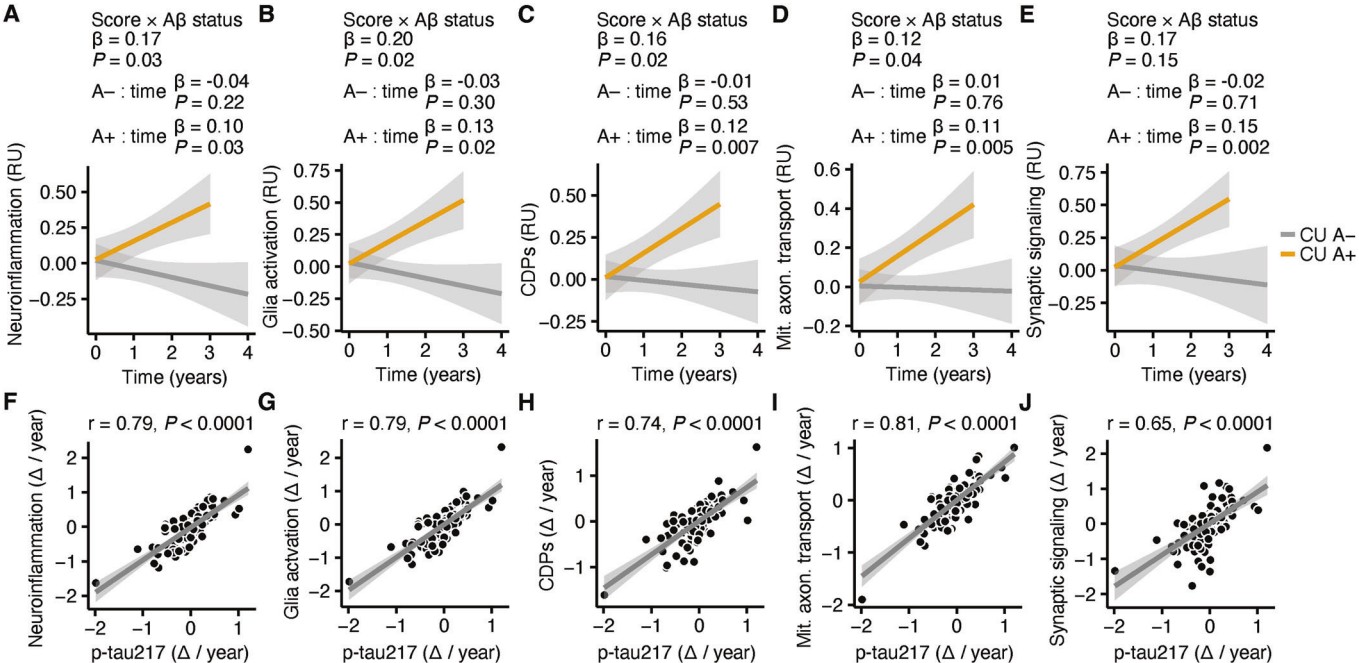

**Figure 4. Glia activation, neuroinflammation, and cell death pathways are associated with pathophysiologic progression in preclinical AD.**

(A–E) Longitudinal change of CSF signatures for "Neuroinflammation" (A), "Glia activation" (B), "Cell death pathways" (CDPs) (C), "Mitochondrial axonal transport" (D), and "Synaptic signaling" (E) in cognitively unimpaired with PET proven Aβ accumulation (CU A+, n = 22), and without Aβ accumulation (CU A−, n = 84). Linear mixed-effect models were used for statistical comparisons. Age, sex at birth, and baseline values of the respective signatures were used as covariates. The data were normalized to the first visit. The interaction between the signatures and amyloid status over time (years), the longitudinal increase in A+ and in A− were tested using mixed linear models. The β-values and P values are shown in the figure. (F–J) Spearman correlation analysis between the change per year of the CSF signatures for "Neuroinflammation" (F), "Glia activation" (G), "Cell death pathways" (CDPs) (H), "Mitochondrial axonal transport" (I), and "Synaptic signaling" (J) and CSF p-tau217 change per year in all cognitively unimpaired (n = 106). Source data are available online for this figure.

activated in early AD. Most CDP research has focused on neurons in the past (Koper et al, 2024; Balusu et al, 2023; Chauhan et al, 2020; LaFerla et al, 1995), but more recent findings in mice and humanized cell culture models suggest an important role for their activation in microglia (Haney et al, 2024; Ryan et al, 2023) and oligodendrocytes (Depp et al, 2023; Blanchard et al, 2022) in AD progression. Intriguingly, activation of pyroptosis, an inflammatory CDP, in microglia leads to the secretion of ACS-specks that strongly promote Aβ accumulation (Heneka, 2017; Venegas et al, 2017), linking inflammation to the earliest AD stages. Further research is required to clarify which cell types are affected by CDPs activation in early AD, to identify biomarkers for specific PCDs, and how this might be therapeutically targeted.

We found that the effect of neuroinflammation on CDP activation in early AD was significantly mediated by abnormal synaptic signaling in both cohorts and axonal mitochondrial transport deficits in TRIAD only, which are among the pathways showing very early involvement across multiple AD cohorts (Pichet Binette et al, 2024). Thus, our data support an interaction between glia-mediated inflammation, neuronal pathologies, and activation of CDPs. This is in line with several studies that demonstrated the interplay between astrocytes of microglia as a decisive regulator of neuronal demise in AD (McAlpine et al, 2021; Liddelow et al, 2017; Guttenplan et al, 2021). Furthermore, our data suggests that cytokine-mediated dysfunction of synaptic signaling and mitochondrial axonal transport contribute to neuronal injury in early

AD. This resemblance to inflammation-induced neurodegeneration in primary inflammatory disease of the CNS like multiple sclerosis (Woo et al, 2024b; Sorbara et al, 2014; Jafari et al, 2021; Woo et al, 2024a, 2024c) or viral CNS inflammation (Di Liberto et al, 2018, 2022) needs to be further causally interrogated in AD mouse and cell culture models.

By interrogating multiple cellular disease pathways in preclinical and early AD, this approach might offer new explanations for the heterogeneity of AD progression and has the potential to pave the way for individualized therapeutic strategies. Similar approaches have been successfully applied in numerous oncological diseases where "liquid biopsies" are used for diagnosis, individualized treatments, and treatment monitoring (Ma et al, 2024; Pantel and Alix-Panabières, 2019). Since the first single-cell sequencing postmortem studies revealed that microglia-mediated phagocytosis is an important predictor for Aβ removal by Aβ-targeting immune strategies (van Olst et al, 2025) and accumulating evidence suggests an important role of inflammation for adverse events like Aβ-related imaging abnormalities (Piazza et al, 2022), the identification of multi-protein inflammatory CSF signatures might be able to inform about therapy response and adverse events.

While we identified several dysregulated pathways in preclinical AD, several methodological limitations remain. Although similar pathways have been described in preclinical individuals with ADAD (Shen et al, 2024), and we have replicated the cross-sectional analyses in two cohorts, longitudinal multiplexed

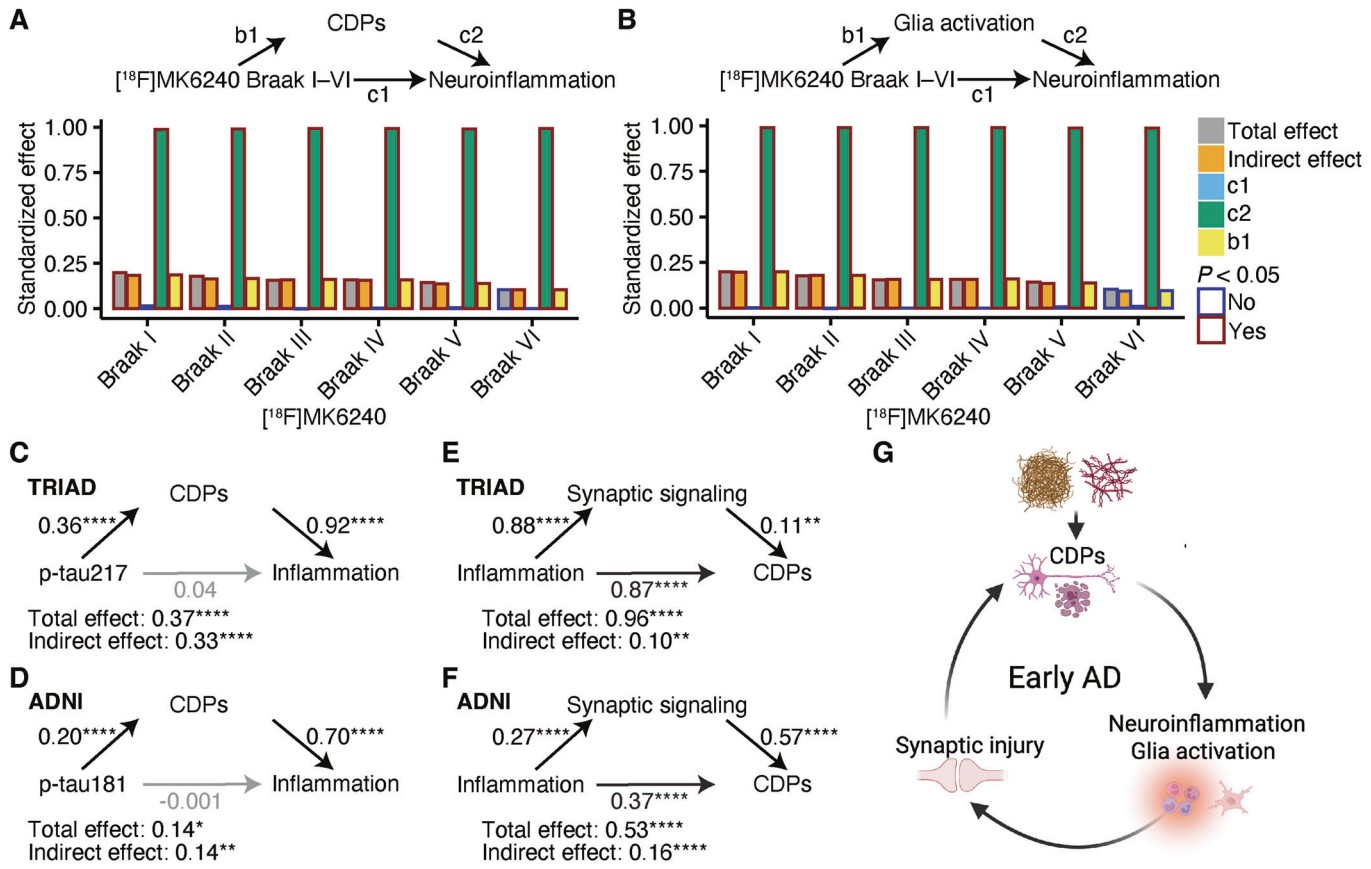

**Figure 5.  Early glia and neuronal dysfunction sustain neuroinflammation in early AD.**

(A, B) Mediation analysis in A-T- and A + T- (n = 204) using [¹⁸F]MK6240 SUVR in Braak I-VI as predictors, neuroinflammation as predicted variable, and cell death pathways (CDPs) (A) and glia activation (B) as mediators in TRIAD. The standardized effects for all paths, as well as the total and indirect effects, are shown. Significance is indicated by coloring. (C, D) Mediation analysis in A-T- and A + T- in TRIAD (C, n = 204), and ADNI (D, n = 145) using CSF p-tau217 as predictor in TRIAD and CSF p-tau181 as predictor in ADNI, neuroinflammation as predicted variable, and CDPs as mediator. The standardized effects for all paths, as well as the total and indirect effects, are shown. ****$P < 0.0001$. (E, F) Mediation analysis in A-T- and A + T- in TRIAD (E, n = 204), and ADNI (F, n = 145) using neuroinflammation as predictor, CDPs as predicted variable, and synaptic signaling as mediator. The standardized effects for all paths, as well as the total and indirect effects, are shown. *$P < 0.05$, **$P < 0.01$, ****$P < 0.0001$. (G) Graphical summary of this study. Neuroinflammation is associated with disease progression in early AD is maintained by glia activation and neuronal pathologies. Source data are available online for this figure.

immunoassay data were only available for TRIAD. Replication of our longitudinal analyses in an independent cohort is required to further demonstrate the association between the identified disease pathways and progression. Second, the CSF proteome reflects brain-intrinsic processes associated with in vivo longitudinal AD progression, but of course, it is not possible to confirm this directly in the context of premortem studies. Therefore, subsequent studies need to focus on PET studies with tracers that directly visualize the inflammatory disease pathologies in vivo (Masdeu et al, 2022). Third, while we also used the 7K SomaScan assay from ADNI, the NULISA CNS panel used in TRIAD is a targeted multiplexed immunoassay approach and therefore targets only a limited subset of CSF proteins. Still, due to its high sensitivity, it may detect changes that are not observed by other platforms (Feng et al, 2023) and therefore, allows the discovery of new pathways, especially in preclinical AD. However, this is biased by the currently available panel, and therefore, additional disease pathways need to be investigated in subsequent studies when additional panels or other technologies with similar sensitivities are available. Although we

replicated key findings with the SomaScan technology that relies on aptamer-based proteomics in contrast to NULISA, that uses proximity ligation approach, assay-specific biases for proteins cannot be fully excluded. Since aptamer-based and proximity ligation-based protein quantifications show significant differences (Eldjarn et al, 2023), large-scale and systematic comparisons between different technologies (Imam et al, 2025) are required to further improve the interpretation of proteomics data from different platforms. Last, our definition of pathways relied on gene ontology pathways. While this is one of the gold standards for pathway analyses, these are restricted to the currently available knowledge., This did not allow a differentiation of glia subtypes and states and resulted in a high overlap of proteins in the glia activation and neuroinflammation pathways. Further studies are required that correlate cell type-specific expression with secretomics to aid in the interpretation of proteomics in body fluids.

The results of our study further underscore the importance of early interventions, given the magnitude of pathophysiological events beyond amyloid and tau aggregates. The results suggest that

a vicious cycle of neuroinflammation, glia activation, and CDPs is associated with disease progression in early and preclinical AD. The identified biomarker candidates may prove useful in combination with established ATN biomarkers to help individualize treatment strategies in the future.

# Methods

**Reagents and tools table**

| Reagent/ resource | Reference or source | Identifier or catalog number |
| --- | --- | --- |
| **Experimental models** | | |
| NA | | |
| **Recombinant DNA** | | |
| NA | | |
| **Antibodies** | | |
| NA | | |
| **Oligonucleotides and other sequence-based reagents** | | |
| NA | | |
| **Chemicals, enzymes, and other reagents** | | |
| NA | | |
| **Software** | | |
| Adobe Illustrator | Adobe Systems | RRID:SCR_010279 |
| clusterProfiler | (Yu et al, 2012) | RRID:SCR_016884 |
| FreeSurfer | (Fischl, 2012) | RRID:SCR_001847 |
| lavaan | (Rosseel, 2012) | NA |
| lme4 | (Robin et al, 2011) | RRID:SCR_000432 |
| R | R Project for Statistical Computing | RRID:SCR_001905 |
| R Studio | R Studio | RRID:SCR_000432 |
| tidyplots | (Engler, 2025) | NA |
| tidyverse | (Wickham et al, 2019) | RRID:SCR_019186 |
| **Other** | | |
| NA | | |

## Methods and protocols

### Study population

We studied participants from the TRIAD cohort (Therriault et al, 2020), who were recruited from the patients of the McGill Centre for Studies in Aging Memory Clinic. CU participants had a Clinical Dementia Rating (CDR) score of 0, no objective cognitive impairment, and preserved activities of daily living (ADL). CI participants included those diagnosed with mild cognitive impairment (MCI) or dementia, with or without biomarker evidence of AD pathology. Participants with MCI had a CDR score of 0.5 and relatively preserved ADL, while those with dementia had CDR scores of 1 or 2 and impaired ADL performance. Further descriptions of the TRIAD cohort, as well as inclusion and

exclusion criteria, are published elsewhere (Therriault et al, 2020). The TRIAD study received approval from the Montreal Neurological Institute (MNI) PET Working Committee and the Douglas Mental Health University Institute Research Ethics Board. All participants provided informed written consent.

All participants underwent extensive clinical evaluations, including comprehensive neuropsychological testing and neurological assessment. To index clinical severity, we used CDR-SOB and MMSE scores. All participants included in this study underwent tau-PET imaging with [18F]MK6240 as well as amyloid-PET with [18F]AZD4694 imaging. As a measure of neurodegeneration, hippocampal volume was estimated with FreeSurfer version 7.4.1 (Fischl, 2012). The CSF measurements are described below in more detail.

Furthermore, we included cognitively unimpaired, preclinical, and AD subjects from the Alzheimer's Disease Neuroimaging Initiative (ADNI) database (adni.loni.usc.edu). The ADNI was launched in 2003 as a public-private partnership, led by Principal Investigator Michael W. Weiner, MD. The primary goal of ADNI has been to test whether serial magnetic resonance imaging (MRI), PET, other biological markers, and clinical and neuropsychological assessment can be combined to measure the progression of MCI and ADD. For up-to-date information, see www.adni-info.org. The study was approved by the Institutional Review Boards of all the participating institutions, and informed written consent was obtained from all participants. The data used for the analyses presented here was accessed on August 1, 2025. We included participants for whom [18F]Florbetapir PET or CSF Aβ1–42, p-tau181 in the CSF, and 7K SomaScan data were available. Tau-PET was not available for any of the individuals. In ADNI, lumbar puncture was performed as described in the ADNI procedures manual (http://www.adni-info.org/). CSF samples were frozen on dry ice within 1 h after collection and shipped overnight on dry ice to the ADNI Biomarker Core laboratory at the University of Pennsylvania Medical Center. CSF Aβ1–42 was measured using the Elecsys β-amyloid(1–42) assay (Bittner et al, 2016).

### NULISA analysis
NULISA assays of the CSF were performed blinded at Alamar Biosciences as described previously using the "CNS Disease Panel 120" (Feng et al, 2023).

### 7K SomaScan analysis
The detailed methodology can be found in the ADNI documentation (adni.loni.usc.edu) and the original publication (Ali et al, 2025). We used the relative fluorescence units after quality control. No longitudinal SomaScan data were available.

### Neuroimaging acquisition and processing
We used a 3T Siemens Magnetom scanner with a standard head coil to obtain T1-weighted structural brain magnetic resonance imaging (MRI) with 1 mm isotropic voxels. T1-weighted MRI scans were processed using a locally developed pipeline, which involved linear and nonlinear registration using ANTS and brain segmentation using FreeSurfer. A brain-dedicated Siemens high-resolution research tomograph was used for PET imaging. [18F]MK6240 and [18F]AZD4694 PET images were obtained over a 90–110 min and 40–70 min interval after intravenous bolus ligand injection, respectively. PET scans were reconstructed with a sequential subset

expectation-maximization algorithm on a 4D volume with four (4 × 300 s for [18F]MK6240) or three (3 × 600 s for [18F]AZD4694) frames. Attenuation, motion, dead time, decay, random, and scattered coincidences corrections were applied. PET images were registered linearly to each subject's T1-weighted space, as well as linearly and nonlinearly to the MNI reference space. A spatial smoothing was applied using a filter with a full width at half maximum of 8 mm. To avoid meningeal off-target binding spillover, we stripped meninges from PET images prior to transformations and blurring (Pascoal et al, 2020). We calculated the standardized value uptake ratio (SUVR) as a proxy for tau load in previously described Braak-like ROIs representing paired hierarchical and cumulative stages of tau accumulation (Macedo et al, 2024a, 2024b). SUVR for tau-PET was calculated using the inferior cerebellar gray matter as the reference region. [18F] AZD4694 SUVRs were calculated using the whole cerebellum gray matter as the reference region.

Detailed descriptions of ADNI neuroimaging acquisition and preprocessing can be found elsewhere (http://adni.loni.usc.edu/datasamples/pet/). We used the neocortical [18F]Florbetapir SUVR that was normalized to the whole cerebellum. We only included data that passed the quality control.

### Biomarker status and staging

Participants were classified according to the amyloid/tau/neurodegeneration (A/T/N) framework by measuring temporal meta-ROI [18F] MK6240 SUVR (cut-off 1.24), neocortical [18F]AZD4694 SUVR (cut-off 1.55), and hippocampal volume by structural MRI (cut-off 3.2 cm³) as previously described (Woo et al, 2024e, 2024d). For individuals where [18F]AZD4694 was not available, we used the CSF Aβ42/Aβ40 ratio from the NULISA dataset. Aβ-positivity was determined as an Aβ42/Aβ40 ratio below 2 SD of the Aβ42/Aβ40 ratio of CUY individuals. Tau-PET positivity for each Braak stage was defined as [18F]MK6240 SUVR >2.5 standard deviations of the mean of CU A− participants, as previously published (Therriault et al, 2022).

For ADNI participants, we used published neocortical [18F] Florbetapir (1.08) (Landau et al, 2013) or if no amyloid-PET was available, CSF Aβ1–42 (981 pg/mL) (Dumurgier et al, 2022) cut-offs for amyloid-positivity and a published CSF p-tau181 (23 pg/mL) (Tan et al, 2020) cut-off for tau-positivity. For N-status, we used hippocampal volume loss >2.5 standard deviations of the mean of the 20 youngest, CU A− ADNI participants.

### Statistical analysis

All analyses were performed with R (v4.4.1). The *tidyverse* and *tidyplots* (Engler, 2025) packages were used for visualizations. We used the NPX NULISA values and replaced all values below the limit of detection with 0. All values higher or lower than 3 standard deviations of all included individuals per protein were excluded as outliers. Z-scores for each protein were calculated using CU A− as a reference. For SomaScan, we used the provided normalized data after QC. We generated z-sores for each protein using CU A− as a reference. If not indicated otherwise, we used unpaired *t*-tests for group comparisons and Spearman correlation analysis. The statistics are specified in the respective figure legends. Significant results are indicated by *$P < 0.05$, **$P < 0.01$, ***$P < 0.001$, and ****$P < 0.0001$.

### Summary scores

We analyzed intraindividual differences and assessed how the biological processes are associated with disease progression.

Therefore, we calculated scores that reflect different biological themes and were composed of different proteins for each individual. These biological themes were defined by gene ontology biological process analyses. We performed AUCell analysis (Aibar et al, 2017) using the proteins of the respective GO terms and generated a z-score using CU A- as references. Individuals with a z-score >5 were excluded as outliers.

### Longitudinal analyses

To identify whether the changes in the different biological themes were associated with AD progression, we performed longitudinal analyses. We calculated the rate of change per year of the z-scores of all our measurements starting from the baseline measurements. We separated CU and CI individuals and calculated the interaction between Aβ status and time using linear mixed effects models with the *lme4* package (Bates et al, 2015). Age, sex, and baseline values were used as covariates, subjects as random intercepts. The *P* value and β-values for the interaction between time and Aβ status are reported in the respective figures.

### Differential abundance analysis of protein levels

To test the significance of protein concentration differences between CUY and other diagnostic groups, we performed two-tailed *t*-tests with false discovery rate (FDR) correction for multiple comparisons. To identify biological themes that summarize several proteins, we performed biological process GO term analysis using *clusterProfiler* (Yu et al, 2012) with the proteins showing the most important concentration changes in the respective comparisons. GO terms that included "migration", "chemotaxis", "ERK-signaling", "Synaptic signaling", "Calcium signaling", "Glia", or "Cytokine" were summarized in the respective groups. We calculated the −log10 of the adjusted *P* value to visualize the significant enrichments.

### Unbiased clustering analysis

To identify protein groups that were associated with AD hallmarks, we calculated the Spearman correlation between each protein and [18F]MK6240 SUVR in the different Braak stages and meta-ROI, Braak stage classification as a numeric variable, neocortical [18F] AZD4694 SUVR, hippocampal volume, CDR-SOB, and MMSE. Next, we used hierarchical clustering and determined the ideal number of clusters by using a dendrogram and an elbow plot. A dissimilarity matrix was used to visualize the differences between the clusters. To characterize the proteins in the different clusters, we performed GO term analysis with the proteins that were included in the respective clusters.

### Structural equation models

To analyze the direct and indirect effects of the biological themes on AD progression, we used structural equation models. The structural equation models were performed using *lavaan* (Rosseel, 2012). We separately analyzed A−T−, A+T− (as early AD), and A+T+. Age, sex, and years of education were used as covariates. The predictors, outcome variables and mediators, as well as the direct and indirect effects were shown in the respective figures. The standardized effects were additionally reported. The root mean square error of approximation (RMSEA) was <0.05, and the incremental fit index (0.95) was >0.95 for each model.

## The paper explained

### Problem

While inflammation has been recognized as an important driver of Alzheimer's disease dementia, its role in preclinical and early Alzheimer's disease remains less explored.

### Results

Biomarkers of glia inflammation and activation of cell death pathways were increasing in amyloid-β positive individuals before any cognitive deficits appeared. Mediation analyses revealed a vicious cycle of glia inflammation and activation of cell death pathways that drove disease progression in preclinical and early Alzheimer's disease.

### Impact

These findings position glia inflammation as a central driver of Alzheimer's disease progression in preclinical and early disease stages, supporting inflammation-focused biomarker and treatment strategies in preclinical and early Alzheimer's disease.

## Graphics

Figure 5G and the synopsis were created with BioRender.com.

## Data availability

All requests for raw and analyzed data and materials will be promptly reviewed by McGill University to verify if the request is subject to any intellectual property or confidentiality obligations. Anonymized data will be shared upon request to the study's senior author from a qualified academic investigator. Any data and materials that can be shared will be released via a material transfer agreement. Data were not publicly available due to information that could compromise the privacy of research participants. Related documents, including study protocol and informed consent forms, can similarly be made available upon request. Data and/or samples from TRIAD can be requested on https://triad.tnl-mcgill.com/contact-us/. The code is available on GitHub: https://github.com/TNL-MCSA/Code-and-material-for-publications/tree/main/2025_EmboMolMed_v1.

The source data of this paper are collected in the following database record: biostudies:S-SCDT-10_1038-S44321-025-00316-1.

## Peer review information

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

## Acknowledgements

We thank the study participants and the McGill Center for Studies in Aging staff. Data used in preparation of this article were obtained from the Alzheimer's Disease Neuroimaging Initiative (ADNI) database (adni.loni.usc.edu). As such, the investigators within the ADNI contributed to the design and implementation of ADNI and/or provided data but did not participate in the analysis or writing of this report. A complete listing of ADNI investigators can be found at: http://adni.loni.usc.edu/wp-content/uploads/how_to_apply/ADNI_Acknowledgement_List.pdf. We thank the ADNI participants and their families who made this study possible. This study was supported by the Canadian Institutes of Health Research (CIHR) [MOP-11-51-31; RFN 152985, 159815, 162303], Canadian Consortium of Neurodegeneration and Aging (CCNA; MOP-11-51-31 -team 1), Weston Brain Institute, the Alzheimer's Association [NIRG-12-92090, NIRP-12-259245], Brain Canada Foundation (CFI Project 34874; 33397), and the Fonds de Recherche du Québec – Santé (FRQS; Chercheur Boursier, 2020-VICO-279314). ACM is supported by the Mitacs Graduate Fellowship (IT27627), the Max E Binz Fellowship-Medicine (F225864C02), and the Grad Excellence Award in Neurology & Neurosurgery (M159875C51). MSW is funded by the Else-Kröner-Fresenius foundation (2023_EKMS.03), the Corona Foundation (S0199/10110/2025), and the German Research Foundation (WO 2835/1-1). The funding

sources had no participation in the design of the study, in the collection, analysis, and interpretation of data, or in the manuscript writing. HZ is a Wallenberg Scholar and a Distinguished Professor at the Swedish Research Council supported by grants from the Swedish Research Council (#2023-00356, #2022-01018, and #2019-02397), the European Union's Horizon Europe research and innovation program under grant agreement No 101053962, Swedish State Support for Clinical Research (#ALFGBG-71320), the Alzheimer Drug Discovery Foundation (ADDF), USA (#201809-2016862), the AD Strategic Fund and the Alzheimer's Association (#ADSF-21-831376-C, #ADSF-21-831381-C, #ADSF-21-831377-C, and #ADSF-24-1284328-C), the European Partnership on Metrology, co-financed from the European Union's Horizon Europe Research and Innovation Program and by the Participating States (NEuroBioStand, #22HLT07), the Bluefield Project, Cure Alzheimer's Fund, the Olav Thon Foundation, the Erling-Persson Family Foundation, Familjen Rönströms Stiftelse, Stiftelsen för Gamla Tjänarinnor, Hjärnfonden, Sweden (#FO2022-0270), the European Union's Horizon 2020 research and innovation program under the Marie Skłodowska-Curie grant agreement No 860197 (MIRIADE), the European Union Joint Program – Neurodegenerative Disease Research (JPND2021-00694), the National Institute for Health and Care Research University College London Hospitals Biomedical Research Centre, the UK Dementia Research Institute at UCL (UKDRI-1003), and an anonymous donor. KB is supported by the Swedish Research Council (#2017-00915 and #2022-00732), the Swedish Alzheimer Foundation (#AF-930351, #AF-939721, #AF-968270, and #AF-994551), Hjärnfonden, Sweden (#ALZ2022-0006, #FO2024-0048-TK-130 and FO2024-0048-HK-24), the Swedish state under the agreement between the Swedish government and the County Councils, the ALF-agreement (#ALFGBG-965240 and #ALFGBG-1006418), the European Union Joint Program for Neurodegenerative Disorders (JPND2019-466-236), the Alzheimer's Association 2021 Zenith Award (ZEN-21-848495), the Alzheimer's Association 2022-2025 Grant (SG-23-1038904 QC), La Fondation Recherche Alzheimer (FRA), Paris, France, the Kirsten and Freddy Johansen Foundation, Copenhagen, Denmark, Familjen Rönströms Stiftelse, Stockholm, Sweden, and an anonymous filantropist and donor.

## Author contributions

**Marcel S Woo**: Conceptualization; Data curation; Formal analysis; Supervision; Investigation; Visualization; Methodology; Writing—original draft; Writing—review and editing. **Joseph Therriault**: Data curation; Writing—review and editing. **Seyyed, Ali Hosseini**: Data curation; Writing—review and editing. **Yi-Ting Wang**: Data curation; Writing—review and editing. **Arthur C Macedo**: Data curation; Writing—review and editing. **Nesrine Rahmouni**: Data curation; Project administration; Writing—review and editing. **Étienne Aumont**: Data curation; Writing—review and editing. **Stijn Servaes**: Data curation; Writing—review and editing. **Cécile Tissot**: Data curation; Writing—review and editing. **Jaime Fernandez-Arias**: Data curation; Writing—review and editing. **Lydia Trudel**: Data curation; Writing—review and editing. **Brandon Hall**: Data curation; Writing—review and editing. **Gleb Bezgin**: Data curation; Writing—review and editing. **Kely Quispialaya-Socualaya**: Data curation; Writing—review and editing. **Marina Goncalves**: Data curation; Writing—review and editing. **Tevy Chan**: Data curation; Writing—review and editing. **Jenna Stevenson**: Data curation; Writing—review and editing. **Yansheng Zheng**: Data curation; Writing—review and editing. **Stuart Mitchell**: Data curation; Writing—review and editing. **Robert Hopewell**: Data curation; Writing—review and editing. **Ilaria Pola**: Data curation; Writing—review and editing. **Kubra Tan**: Data curation; Writing—review and editing. **Guglielmo Di Molfetta**: Data curation; Writing—review and editing. **Firoza Z Lussier**: Data curation; Writing—review and editing. **Gassan Massarweh**: Data curation; Writing—review and editing. **Paolo Vitali**: Data curation; Writing—review and editing. **Jean-Paul Soucy**: Data curation; Writing—review and editing. **Serge Gauthier**: Data curation; Writing—review and editing. **Nicholas J Ashton**: Data curation; Writing—review and editing. **Kaj Blennow**: Data curation; Writing—review and editing. **Tharick A Pascoal**: Data curation; Writing—review and editing. **Henrik Zetterberg**: Data curation; Writing—review and editing. **Andréa L Benedet**: Data curation; Investigation; Writing—original draft; Project administration; Writing—review and editing. **Pedro Rosa-Neto**: Conceptualization; Resources; Data curation; Formal analysis; Supervision; Funding acquisition; Investigation; Writing—original draft; Project administration; Writing—review and editing.

Source data underlying figure panels in this paper may have individual authorship assigned. Where available, figure panel/source data authorship is listed in the following database record: biostudies:S-SCDT-10_1038-S44321-025-00316-1.

## Funding

## Disclosure and competing interests statement

MSW receives honoraria from Lilly for educational lectures outside the scope of this work. HZ has served at scientific advisory boards and/or as a consultant for Abbvie, Acumen, Alector, Alzinova, ALZpath, Amylyx, Annexon, Apellis, Artery Therapeutics, AZTherapies, Cognito Therapeutics, CogRx, Denali, Eisai, Enigma, LabCorp, Merry Life, Nervgen, Novo Nordisk, Optoceutics, Passage Bio, Pinteon Therapeutics, Prothena, Quanterix, Red Abbey Labs, reMYND, Roche, Samumed, Siemens Healthineers, Triplet Therapeutics, and Wave, has given lectures sponsored by Alzecure, BioArctic, Biogen, Cellectricon, Fujirebio, Lilly, Novo Nordisk, Roche, and WebMD, and is a co-founder of Brain Biomarker Solutions in Gothenburg AB (BBS), which is a part of the GU Ventures Incubator Program (outside submitted work). KB has served as a consultant and on advisory boards for AbbVie, AC Immune, ALZpath, AriBio, Beckman-Coulter, BioArctic, Biogen, Eisai, Lilly, Moleac Pte. Ltd, Neurimmune, Novartis, Ono Pharma, Prothena, Quanterix, Roche Diagnostics, Sanofi, and Siemens Healthineers; has served at data monitoring committees for Julius Clinical and Novartis; has given lectures, produced educational materials and participated in educational programs for AC Immune, Biogen, Celdara Medical, Eisai and Roche Diagnostics; and is a co-founder of Brain Biomarker Solutions in Gothenburg AB (BBS), which is a part of the GU Ventures Incubator Program, outside the work presented in this paper.

