## [Peer Review File · EMBO Molecular Medicine]

Glia inflammation and cell death pathways drive disease progression in preclinical and early AD

Marcel Woo, Joseph Therriault, Seyyed Hosseini, Yi-Ting Wang, Arthur Macedo, Nesrine Rahmouni, Etienne Aumont, Stijn Servaes, Cecile Tissot, Jaime Fernandez Arias, Lydia Trudel, Brandon Hall, Gleb Bezgin, Kely Quispialaya-Socualaya, Marina Goncalves, Tevy Chan, Jenna Stevenson, Yansheng Zheng, Stuart Mitchell, Robert Hopewell, Ilaria Pola, Kubra Tan, Guglielmo Di Molfetta, Firoza Lussier, Gassan Massarweh, Paolo Vitali, Jean-Paul Soucy, Serge Gauthier, Nicholas Ashton, Kaj Blennow, Tharick A. Pascoal, Henrik Zetterberg, Andréa L. Benedet, and Pedro Rosa-Neto

Corresponding authors: Marcel Woo (m.woo@uke.de) , Pedro Rosa-Neto (pedro.rosa@mcgill.ca)

Review Timeline:

Submission Date:	17th Jun 25
Editorial Decision:	28th Jul 25
Revision Received:	17th Aug 25
Editorial Decision:	9th Sep 25
Revision Received:	18th Sep 25
Accepted:	19th Sep 25

Editor: Jingyi Hou

Transaction Report:

29th Jul 2025

Dear Dr. Woo,

Thank you again for submitting your work to EMBO Molecular Medicine. We have now received the reports from the two reviewers and as you will see below, the reviewers think that the study is potentially interesting. They raise however a series of concerns, which we would ask you to convincingly address in a revision.

I think the referees' recommendations are clear and need not be repeated here. In particular, Reviewer #2 noted that the conclusions would be strengthened by replicating key findings in an independent cohort or using publicly available datasets. We strongly encourage you to address this point.

As you may already know, our editorial policy allows in principle a single round of major revision so it is essential to provide responses to the referees' comments that are as complete as possible. Please feel free to contact me in case you would like to discuss in further detail any of the issues raised by the referees.

Please also contact us as soon as possible if similar work is published elsewhere. If other work is published, we may not be able to extend the revision period beyond three months.

I look forward to receiving your revised manuscript.

Yours sincerely,
Jingyi

Jingyi Hou
Senior Editor
EMBO Molecular Medicine

We require:

2) Individual production quality figure files as .eps, .tif, .jpg (one file per figure). For guidance, download the 'Figure Guide PDF': (<https://www.embopress.org/page/journal/17574684/authorguide#figureformat>).

3) A .docx formatted letter INCLUDING the reviewers' reports and your detailed point-by-point responses to their comments. As part of the EMBO Press transparent editorial process, the point-by-point response is part of the Review Process File (RPF), which will be published alongside your paper.

4) A complete author checklist, which you can download from our author guidelines (<https://www.embopress.org/page/journal/17574684/authorguide#submissionofrevisions>). Please insert information in the checklist that is also reflected in the manuscript. The completed author checklist will also be part of the RPF.

6) It is mandatory to include a 'Data Availability' section after the Materials and Methods. Before submitting your revision, primary datasets produced in this study need to be deposited in an appropriate public database, and the accession numbers and database listed under 'Data Availability'. Please remember to provide a reviewer password if the datasets are not yet public (see <https://www.embopress.org/page/journal/17574684/authorguide#dataavailability>).

12) Author contributions: You will be asked to provide CRediT (Contributor Role Taxonomy) terms in the submission system. These replace a narrative author contribution section in the manuscript.

13) A Conflict of Interest statement should be provided in the main text.

14) Please provide a 'Synopsis' to further enhance discoverability. Synopses are displayed on the journal webpage and are

freely accessible to all readers. They include a short stand first (maximum of 300 characters, including space) as well as 2-5 one-sentences bullet points that summarizes the paper. Please write the bullet points to summarize the key NEW findings. They should be designed to be complementary to the abstract - i.e. not repeat the same text. We encourage inclusion of key acronyms and quantitative information (maximum of 30 words / bullet point). Please use the passive voice. Please attach these in a separate file or send them by email, we will incorporate them accordingly.

Please also provide a visual abstract to illustrate your article as a PNG file 550 px wide x 300-600 px high.

15) All Materials and Methods need to be described in the main text using our 'Structured Methods' format. According to this format, the Methods section includes a Reagents and Tools Table (listing key reagents, experimental models, software and relevant equipment and including their sources and relevant identifiers) followed by a Methods and Protocols section describing the methods, ideally using a step-by-step protocol format. The aim is to facilitate adoption of the methodologies across labs.

Please download and fill our Reagents and Tools Table template (.docx), which you can find in our author guidelines: <https://www.embopress.org/page/journal/17574684/authorguide#structuredmethods>

**** Reviewer's comments ****

Referee #1 (Remarks for Author):

This ms by Woo and colleagues adds on the increasing knowledge about the role of neuroinflammation in neurodegenerative disease and in particular Alzheimer's. While the topic is timely and of interest to the readers of this journal, there are several shortcomings which need to be further addressed as listed below.

General: The experimental approach is more or less a fishing exercise which is biased by the assays own restriction, meaning that only those targets can be detected, which are being covered by the respective panel. It is thus, far from giving a complete picture of protein and peptide level changes, a critical discuss on this is entirely missing. Likewise, influencing factors such as protein turnover, half-life times and source are hardly addressed.

Detailed critique

Figure 1c: How are these annotations defined, what are they based on?, on a technical note, the individual pathways are hard to read and the font should be increased. Alternatively, these could be given in a table. Figure 1e: How do the authors define "glia activation", how do they distinguish changes present in astrocytes, microglia, boarder macrophages ? In keeping with this, it would be interesting to learn whether "glia activation" involves anti- and pro inflammatory changes.

Figure 2a: how do the authors explain that tau species measured (tau 181, tau 231 and tau 217 one weakly correlate with hippocampal volume? Figure 2D how are "positive regulation of inflammatory response", "glia activation" and "regulation of inflammatory responses" being distinguished? By which logic are these being separated?

Figure 3 Lacks novelty. How do the authors distinguish A:B and E:F "Glia activation" from "Neuroinflammation". This seems to this reviewer entirely artificial and factors that influence this differentiation remain unclear to the reader. Early upregulation of inflammatory signaling cascades have been described at large and need to be referenced more appropriately!

Figure 4 See comments above. Again it is difficult to understand the way Glia activation and Neuroinflammation are being distinguished. Both are part of the initial and sustained DAMP reaction. Given the fact that tau species have been identified as strong DAMP it seems interesting and surprising that at the occurrence of tau pathology (A+T-N- vs A+T+N-) does not increase Neuroinflammation? Please comment.

Figure 5 Regarding Neuroinflammation/Glia activation same comment as before. Beyond this, how to the authors explain the "decrease of Neuroinflammation" and "Glia activation" in the cognitively unimpaired A- population?

Referee #2 (Remarks for Author):

This study by Woo et al. presents an investigation into the biological mechanisms underlying early Alzheimer's disease (AD)

progression. Using CSF proteomics, PET/MRI imaging, and longitudinal data from a well-characterized cohort of 382 individuals, the authors demonstrate the central role of neuroinflammation and neuronal injury in driving AD pathogenesis and identify glia-mediated inflammation and cell death pathways (CDPs) as early drivers of AD progression, particularly in A β + cognitively unimpaired individuals. Overall, this study integrates multiple modalities with a range of sophisticated techniques and robust statistical methods to enhance the interpretability and causal inference of the results. These findings offer a compelling and new insight into AD pathophysiology. The reviewer has a few comments as follows:

1. The authors report no significant GFAP upregulation in CI A+ and CU A+ participants, contrasting with previous plasma/CSF studies. This result is rather strange. While their approach focuses on pathway-level changes, some discussion of platform sensitivity differences or sampling effects would be helpful.
2. Regarding the Results section, can the authors provide more details of the key findings in the manuscript rather than only showing the figure number? It would be more informative and much easier for the audience to capture the key findings.
3. While the TRIAD cohort is well-characterized, the conclusions would be strengthened by replication of key findings in an independent cohort or using publicly available datasets (like ADNI Somascan data, if possible).

Point-to-point response

The changes in the manuscript are referenced in this point-to-point response and are labeled in **red** throughout the manuscript. New figures and tables are marked in **bold**.

Referee #1 (Remarks for Author):

This ms by Woo and colleagues adds on the increasing knowledge about the role of neuroinflammation in neurodegenerative disease and in particular Alzheimer's. While the topic is timely and of interest to the readers of this journal, there are several shortcomings which need to be further addressed as listed below.

General: The experimental approach is more or less a fishing exercise which is biased by the assays own restriction, meaning that only those targets can be detected, which are being covered by the respective panel. It is thus, far from giving a complete picture of protein and peptide level changes, a critical discuss on this is entirely missing. Likewise, influencing factors such as protein turnover, half-life times and source are hardly addressed.

Response: We thank the Reviewer for his overall positive evaluation of our manuscript. The Reviewer raised several important concerns that we addressed with clearer explanations and additional analyses including repeating key results in a second cohort. Addressing the Reviewer's concerns have significantly improved the overall quality of the manuscript.

Detailed critique

Figure 1c: How are these annotations defined, what are they based on?, on a technical note, the individual pathways are hard to read and the font should be increased. Alternatively, these could be given in a table. Figure 1e: How do the authors define "glia activation", how do they distinguish changes present in astrocytes, microglia, boarder macrophages ? In keeping with this, it would be interesting to learn whether "glia activation" involves anti- and pro inflammatory changes.

Response: We agree with the Reviewer that this needs further clarification. The definitions are based on gene ontology biological pathways analyses. This is a well-established database with manually and automatically anotated biological pathways which uses hypergeometric tests to determine signifcant enrichment of proteinsets of interest (Ashburner *et al*, 2000; Aleksander *et al*, 2023). We have now clarified this in the results section. The use of gene ontology terms is limited by the current knowledge and does not clearly allow cell type anotation like for example in techniques like single-cell sequencing. This has been added to the limitations in the discussion.

We apologize for the small fontsize. As suggested, we have now included a list of of the gene ontology biology pathway terms in the new **Appendix Table 3**.

The limited number of NULISA proteins in the CNS panel did not allow us to further differentiate different immune cell or glia cell populations. Similarly, although the GO terms itself include pro- and anti-inflammatory proteins, the limited number of proteins does not allow a clear differentiation of pro- and anti-inflammatory pathways. This is an important limitation which we included in the results and as limitation in the discussion.

Figure 2a: how do the authors explain that tau species measured (tau 181, tau 231 and tau 217 one weakly correlate with hippocampal volume? Figure 2D how are "positive regulation of inflammatory response", "glia activation" and "regulation of inflammatory responses" being distinguished? By which logic are these being separated?

Response: We agree with the Reviewer that this needs clarification. The correlation coefficients for hippocampal volume and cognitive scores were inverted to get positive correlations for more volume loss or more cognitive dysfunction. The correlations for p-tau181, 217, and 231 with hippocampal volume loss were 0.37, 0.36, and 0.35 respectively and were highly significant with adjusted P-values $< 10^{-8}$. These correlations are in the range of published correlations between these CSF p-tau analytes and hippocampal volume (Hempel *et al*, 2005; de Souza *et al*, 2012; Apostolova *et al*, 2010; Mendes *et al*, 2024). However, the bichromatic color scale is misleading and therefore, we changed it to a monochromatic scale in the new **Fig. 2B** to highlight also the significant correlations between the p-tau analytes and hippocampal volume loss.

We agree that this needs further clarification. We performed a gene ontology biological pathway analysis with all proteins in the cluster and used a emaplot for visualizatins. While gene ontology analysis is one of the gold standards, its hierarchical structure with "mother" (like "regulation of inflammatory responses") and subsequent "children" terms (like "positive regulation of inflammatory response" or "glia activation") that contain less proteins and therefore, are more specific. However, this leads to redundant GO term namings. The goal of the visualization was to show a cluster of these similar inflammatory terms. This explanation has been added to the limitations of the discussion. A full list of the top 30 GO terms can be found in the source data.

Figure 3 Lacks novelty. How do the authors distinguish A:B and E:F "Glia activation" from "Neuroinflammation". This seems to this reviewer entirely artificial and factors that influence this differentiation remain unclear to the reader. Early upregulazion of inflammatory signaling cascades have been described at large and need to be referenced more appropriately!

Response: We agree with the Reviewer and therefore, changed this figure to a Appendix Figure.

We have now elaborated on the generation of the scores in the results, and methods section. These terms are based on gene ontology pathways and we generated a composite score using all proteins included in the respective terms. "Neuroinflammation" is a mother-term of "glia activation" and consists of all proteins that are included in the "glia activation" term (plus additional proteins). This clarification has been added to the results. In the TRIAD and ADNI cohort, the "neuroinflammation" and "glia activation" scores are slightly differently associated with AD hallmarks (e.g. Fig. 3A-H), which might suggest a benefit of using more specific pathway definitions. However, we are aware that these terms are limited by the available proteins in the NULISA assay, the currently available knowledge of protein pathways, and by the not allowing for further differentiation into specific pathways are cell types. This is now elaborated as limitations in the discussion. The proteins used for the respective definitions are provided in the new **Appendix Table 4**.

Due to the high overlap of proteins in the "glia activation" and "neuroinflammation" scores, we have now removed the mediation analyses with "neuroinflammation" as predictor and "glia activation" as predicted variable from the new Fig. 5 due to the high expected correlation of the scores. However, we replicated our other mediation analyses in ADNI, and demonstrate in 2 independent cohorts that cell death pathway activation is an important mediator for neuroinflammation in early AD stages. Furthermore, we could also replicated in ADNI that the CSF synapting signalling signature is a significant mediator of the association between neuroinflammation and CDP activation further supporting that the interaction of neuroinflammation, cell death activation and neuronal pathologies are key drivers of early AD. This new data included in the new **Fig. 5D, 5F, Appendix Fig. 6E**.

Furthermore, we have included several studies that describe early inflammation in AD as references. This includes references describing neuroinflammatory biomarkers (Heneka *et al*, 2025; Foley *et al*, 2024; Shen *et al*, 2019), disease mechanisms like pyroptosis in microglia (Heneka, 2017; Venegas *et al*, 2017) and the interplay between glia cells and neurons (Liddelow *et al*, 2017; Guttenplan *et al*, 2021; McAlpine *et al*, 2021), and other available CSF screenings (Ali *et al*, 2025).

Figure 4 See comments above. Again it is difficult to understand the way Glia activation and Neuroinflammation are being distinguished. Both are part of the initial and sustained DAMP reaction. Given the fact that tau species have been identified as strong DAMP it seems interesting and surprising that at the occurrence of tau pathology (A+T-N- vs A+T+N-) does not increase Neuroinflammation? Please comment.

Response: Regarding the Neuroinflammation/Glia activation separation, we kindly refer to 4th response.

We thank the Reviewer for pointing this out. Indeed, tau species are strong DAMPs that contribute to the initial and sustained DAMP reaction. The P-value for the comparison between A+T-N-, and A+T+N- in Fig. 3E (old Fig. 4A) is 0.053 and probably would reach significance after increasing the sample size. The P-value has been added to the figure. Furthermore, we found a significant correlation between the CSF "Neuroinflammation" and "Glia activation" signatures with tau-PET SUVR in all Braak stages in the A-T- and A+T- TRIAD participants (Fig. 3I) supporting that these inflammatory proteins are associated with tau pathology especially in the early disease stages (before they are classified as tau-positive using the tau-PET SUVR in the temporal meta-ROI).

Furthermore, we generated the same scores using the same defining proteins and methods with the 7K SomaScan data from ADNI. Here, we found a significant increase of the CSF "Neuroinflammation" and "Glia activation" signatures in A+T-N- vs. A+T+N-. The differences between TRIAD and ADNI are likely explained by the different strategies for A/T/N classification in the two cohorts since no tau-PET data was available for these ADNI participants. Thus, we show in 2 independent cohorts an association between tau-pathology and CSF signatures of "neuroinflammation" and "glia activation". The new ADNI data is included in **Fig. 3A-D**.

Figure 5 Regarding Neuroinflammation/Glia activation same comment as before. Beyond this, how to the authors explain the "decrease of Neuroinflammation" and "Glia activation" in the cognitively unimpaired A- population?

Response: Regarding the Neuroinflammation/Glia activation separation, we kindly refer to 4th response.

We agree with the Reviewer that this need further clarification. In addition to the interaction analyses between amyloid status and longitudinal progression of the scores, we now also included an interaction analyses between time and the longitudinal progression separately for A+ and A- using mixed linear effect models. This analysis tests whether the scores increase over time in the A+ and A- individuals. This analysis clearly shows a significant increase of these pathways only in A+ but not in A-. Thus, there is no significant difference over time in the A- individuals. We have added these statistical analyses including the P-values and β -coefficients in the new **Fig. 4A-E**.

Referee #2 (Remarks for Author):

This study by Woo et al. presents an investigation into the biological mechanisms underlying early Alzheimer's disease (AD) progression. Using CSF proteomics, PET/MRI imaging, and longitudinal data from a well-characterized cohort of 382 individuals, the authors demonstrate the central role of neuroinflammation and neuronal injury in driving AD pathogenesis and identify glia-mediated inflammation and cell death pathways (CDPs) as early drivers of AD progression, particularly in A β + cognitively unimpaired individuals. Overall, this study integrates multiple modalities with a range of sophisticated techniques and robust statistical methods to enhance the interpretability and causal inference of the results. These findings offer a compelling and new insight into AD pathophysiology. The reviewer has a few comments as follows:

Response: We thank the Reviewer for recognizing the novelty and robustness of our statistical analyses. As suggested, we have now repeated key analyses using the SomaScan data available in ADNI which further strengthens the findings of our manuscript.

1. The authors report no significant GFAP upregulation in CI A+ and CU A+ participants, contrasting with previous plasma/CSF studies. This result is rather strange. While their approach focuses on pathway-level changes, some discussion of platform sensitivity differences or sampling effects would be helpful.

Response: We agree that this needs clarification. Our and other groups have shown in multiple cohorts that GFAP does not increase in A+ individuals in the CSF while it robustly increases in the plasma in CU A+ and CI A+ in comparison to CU A- (Pereira *et al*, 2021; Benedet *et al*, 2021). Thus, our GFAP CSF data is in accordance with the literature. To further show that NULISA reproduces the GFAP findings from the plasma, we now also included from the same individuals plasma GFAP levels measured by NULISA. This shows a robust increase in CU A+ and CI A+ in comparison to CU A-. This new data is now shown in the new **Appendix Fig. 1J-K**. Elucidating the differences between plasma and CSF GFAP levels needs to be subject of future research.

We furthermore agree, that a critical discussion about possible biases due to protein quantification methods is needed. Although, we have replicated key findings using the aptamer-based SomaScan approach, we cannot exclude assay-specific biases of the technologies. Since aptamer-based and proximity ligation-based approaches show significant differences (Eldjarn *et al*, 2023) large scale and systematic comparisons are needed to guide the interpretation of the different proteomics approaches. This has been included in the limitations section.

2. Regarding the Results section, can the authors provide more details of the key findings in the manuscript rather than only showing the figure number? It would be more informative and much easier for the audience to capture the key findings.

Response: We thank the Reviewer for pointing this out. We have now included summary sentences with the key findings after each results section.

3. While the TRIAD cohort is well-characterized, the conclusions would be strengthened by replication of key findings in an independent cohort or using publicly available datasets (like ADNI Somascan data, if possible).

Response: We agree with the Reviewer that replication with a second cohort would add valuable information. As suggested, we used the CSF 7K SomaScan data that is available in ADNI. We included 114 CU, 213 MCI, and 103 ADD patients where A/T/N classification was possible due to availability of amyloid PET, CSF A β 42, CSF p-tau181 and structural MRI. Tau-Pet as well as longitudinal SomaScan data were not available for these individuals

We performed differential abundance analysis between CU A- and CI A+ as well as CU A- and CU A+. This revealed a significant enrichment of similar gene ontology terms like in TRIAD including metabolic regulation, synaptic signalling and glia activation. Additionally, we found a strong upregulation of a protein folding CSF proteomic signature. Thus, we could replicate that these biological pathways are already increased in preclinical AD in group-level comparisons. The new data are included in the new **Figures 1E-H and 1M-P**.

Additionally, we generated scores for "neuroinflammation", "glia activation", "synaptic signalling", "mitochondrial axonal transport", and "cell death pathway activation" using the same proteins and methods as for the NULISA data from TRIAD. We could replicate an increase of these CSF protein signatures in the diagnostic groups and in A/T/N stages further supporting that these pathways are associated with AD hallmarks also in individual-level analyses. The new data are included in the new **Figures 3A-D, Appendix Figures 3A-E, 4B**.

Last, we also replicated the mediation analyses in the A-T- and A+T- ADNI participants. Since p-tau217 was not available for these individuals, we used CSF p-tau181 as progression biomarker. We could replicate that PCD activation mediated the effect of p-tau181 on neuroinflammation, while there was no direct significant effect of p-tau181 on neuroinflammation. Furthermore, we could replicate that synaptic signalling was a significant mediator of the association between neuroinflammation and CDPs. Replication of the mediation analyses further supports that activation of CDPs are an important mediator of early neuroinflammation while neuroinflammation leads to further CDP activation by perturbing synaptic homeostasis. The new data are included in the new **Fig. 5D, 5F, Appendix Fig. 6E**.

References

- Aleksander SA, Balhoff J, Carbon S, Cherry JM, Drabkin HJ, Ebert D, Feuermann M, Gaudet P, Harris NL, Hill DP, *et al* (2023) The Gene Ontology knowledgebase in 2023. *Genetics* 224
- Ali M, Timsina J, Western D, Liu M, Beric A, Budde J, Do A, Heo G, Wang L, Gentsch J, *et al* (2025) Multi-cohort cerebrospinal fluid proteomics identifies robust molecular signatures across the Alzheimer disease continuum. *Neuron* 113: 1363-1379.e9
- Apostolova LG, Hwang KS, Andrawis JP, Green AE, Babakchian S, Morra JH, Cummings JL, Toga AW, Trojanowski JQ, Shaw LM, *et al* (2010) 3D PIB and CSF biomarker associations with hippocampal atrophy in ADNI subjects. *Neurobiol Aging* 31: 1284-1303
- Ashburner M, Ball CA, Blake JA, Botstein D, Butler H, Cherry JM, Davis AP, Dolinski K, Dwight SS, Eppig JT, *et al* (2000) Gene Ontology: tool for the unification of biology. *Nat Genet* 25: 25-29
- Benedet AL, Milà-Alomà M, Vrillon A, Ashton NJ, Pascoal TA, Lussier F, Karikari TK, Hourregue C, Cognat E, Dumurgier J, *et al* (2021) Differences Between Plasma and Cerebrospinal Fluid Glial Fibrillary Acidic Protein Levels Across the Alzheimer Disease Continuum. *JAMA Neurol* 78: 1471
- Eldjarn GH, Ferkingstad E, Lund SH, Helgason H, Magnusson OT, Gunnarsdottir K, Olafsdottir TA, Halldorsson B V., Olason PI, Zink F, *et al* (2023) Large-scale plasma proteomics comparisons through genetics and disease associations. *Nature* 622: 348-358
- Foley KE, Winder Z, Sudduth TL, Martin BJ, Nelson PT, Jicha GA, Harp JP, Weekman EM & Wilcock DM (2024) Alzheimer's disease and inflammatory biomarkers positively correlate in plasma in the UK- ADRC cohort. *Alzheimer's Dement* 20: 1374-1386
- Guttenplan KA, Weigel MK, Prakash P, Wijewardhane PR, Hasel P, Rufen-Blanchette U, Münch AE, Blum JA, Fine J, Neal MC, *et al* (2021) Neurotoxic reactive astrocytes induce cell death via saturated lipids. *Nature* 599: 102-107
- Hempel H, Bürger K, Pruessner JC, Zinkowski R, DeBernardis J, Kerkman D, Leinsinger G, Evans AC, Davies P, Möller H-J, *et al* (2005) Correlation of Cerebrospinal Fluid Levels of Tau Protein Phosphorylated at Threonine 231 With Rates of Hippocampal Atrophy in Alzheimer Disease. *Arch Neurol* 62: 770
- Heneka MT (2017) Inflammation and innate immunity in Alzheimer's disease. *Brain Pathol* 27: 220-222
- Heneka MT, Gauthier S, Chandekar SA, Hviid Hahn-Pedersen J, Bentsen MA & Zetterberg H (2025) Neuroinflammatory fluid biomarkers in patients with Alzheimer's disease: a systematic literature review. *Mol Psychiatry* 30: 2783-2798
- Liddel SA, Guttenplan KA, Clarke LE, Bennett FC, Bohlen CJ, Schirmer L, Bennett ML, Münch AE, Chung W-S, Peterson TC, *et al* (2017) Neurotoxic reactive astrocytes are induced by activated microglia. *Nature* 541: 481-487
- McAlpine CS, Park J, Griciuc A, Kim E, Choi SH, Iwamoto Y, Kiss MG, Christie KA, Vinegoni C, Poller WC, *et al* (2021) Astrocytic interleukin-3 programs microglia and limits Alzheimer's disease. *Nature* 595: 701-706
- Mendes AJ, Ribaldi F, Lathuiliere A, Ashton NJ, Janelidze S, Zetterberg H, Scheffler M, Assal F, Garibotto V, Blennow K, *et al* (2024) Head-to-head study of diagnostic accuracy of plasma and cerebrospinal fluid p-tau217 versus p-tau181 and p-tau231 in a memory clinic cohort. *J Neurol* 271: 2053-2066
- Pereira JB, Janelidze S, Smith R, Mattsson-Carlgen N, Palmqvist S, Teunissen CE, Zetterberg H, Stomrud E, Ashton NJ, Blennow K, *et al* (2021) Plasma GFAP is an early marker of amyloid- β but not tau pathology in Alzheimer's disease. *Brain* 144: 3505-3516
- Shen X-N, Niu L-D, Wang Y-J, Cao X-P, Liu Q, Tan L, Zhang C & Yu J-T (2019) Inflammatory markers in Alzheimer's disease and mild cognitive impairment: a meta-analysis and systematic review of 170 studies. *J Neurol Neurosurg Psychiatry* 90: 590-598
- de Souza LC, Chupin M, Lamari F, Jardel C, Leclercq D, Colliot O, Lehericy S, Dubois B & Sarazin M (2012) CSF tau markers are correlated with hippocampal volume in Alzheimer's disease. *Neurobiol Aging* 33: 1253-1257
- Venegas C, Kumar S, Franklin BS, Dierkes T, Brinkschulte R, Tejera D, Vieira-Saecker A, Schwartz S, Santarelli F, Kummer MP, *et al* (2017) Microglia-derived ASC specks cross-seed amyloid- β in Alzheimer's disease. *Nature* 552: 355-361

9th Sep 2025

Dear Dr. Woo,

Thank you for the submission of your revised manuscript to EMBO Molecular Medicine. We have now received the enclosed report from the referees who were asked to re-assess it. As you will see, the referees are now supportive, and I am pleased to inform you that we will be able to accept your manuscript pending the following amendments:

1. Please provide up to five keywords in the manuscript file.
2. Please remove the "Authors' Contributions" section and "Running title" from the manuscript text.
3. The section titled "Competing interests" should be renamed to "Disclosure statement and competing interests".
4. Authors: for ADNI, please include the full list of members and their affiliations in the Appendix:
<https://www.nature.com/documents/nr-consortia-formatting.pdf>
5. Please ensure consistent use of either 'glial' or 'glia' throughout the manuscript.
6. Funding information needs to be merged with Acknowledgments section. Please verify that the funding details (including project numbers) in the manuscript match those entered in the submission system. The line-by-line entries provided in our system will be automatically linked upon publication, so accuracy and completeness are essential.
7. Appendix: Please add page numbers to the table of contents and remove any red font. The final version of the appendix should be uploaded as a PDF file.
8. Source Data:
 - Please compress the source data for the appendix figures into a single ZIP folder.
 - For the main figures, provide the source data as one file per figure, and name each file using the following format: "[ManuscriptID]_SourceDataForFigureX".
9. Data availability:
 - Please merge the Code Availability section with the Data Availability section. The code should be deposited in an appropriate public repository and made accessible.
 - Please revise the following sentence:
"Anonymized data will be shared upon request to the study's senior author from a qualified academic investigator for the sole purpose of replicating the procedures and results presented in this article"
to:
"Anonymized data will be shared upon request to the study's senior author from a qualified academic investigator."
- The following link is currently not working: <https://datashare.tnl-mcgill.com/ds/>. Please provide a valid, resolvable link.
10. Please address the following issues related to figure legends:
 - Please note that the figure 3G is mislabeled as figure 3D in the manuscript. This needs to be rectified.
 - Please note that the exact p values are not provided in the legends of figures 1C, D, G, H, K, L, O, P; 3A-H; 4F-J.
 - Please indicate the statistical test used for data analysis in the legends of figures 2D, 4A-E.
11. Please remove the information regarding BioRender from the Acknowledgments and add it to the Methods in a dedicated section following this format:
Graphics:
(some of the... OR Figure #... OR synopsis) Graphics were createdwith BioRender.com.
12. Please correct the order and headings of the manuscript sections to: Abstract / Keywords / The Paper Explained / Introduction / Results / Discussion / Methods / Data Availability / Acknowledgements / Disclosure and Competing Interests Statement / References / Main Figure Legends / Tables / Expanded View Figure Legends

I look forward to reading a new revised version of your manuscript as soon as possible.

Sincerely,
Jingyi

Jingyi Hou
Senior Editor
EMBO Molecular Medicine

*** Instructions to submit your revised manuscript ***

***** Reviewer's comments *****

Referee #1 (Remarks for Author):

All my previous concerns have been fully addressed!

Referee #2 (Remarks for Author):

Thanks a lot for the authors providing additional validation analysis based on ADNI CSF SomaScan data. The reviewer has no more comments. Congratulations to the authors' insightful findings.

The authors addressed the remaining editorial issues.

19th Sep 2025

Dear Dr. Woo,

We are pleased to inform you that your manuscript is accepted for publication and is now being sent to our publisher to be included in the next available issue of EMBO Molecular Medicine.

Yours sincerely,
Jingyi

Jingyi Hou
Senior Editor
EMBO Molecular Medicine
